# Integrative and distinctive coding of visual and conceptual object features in the ventral visual stream

Chris B Martin[1]*, Danielle Douglas[2], Rachel N Newsome[3], Louisa LY Man[4], Morgan D Barense[1]*

[1]Department of Psychology, University of Toronto, Toronto, Canada; [2]Department of Psychology, Mount Allison University, Sackville, Canada; [3]Rotman Research Institute, Baycrest, Toronto, Canada; [4]Department of Psychology, Queen's University, Kingston, Canada

**Abstract** A significant body of research in cognitive neuroscience is aimed at understanding how object concepts are represented in the human brain. However, it remains unknown whether and where the visual and abstract conceptual features that define an object concept are integrated. We addressed this issue by comparing the neural pattern similarities among object-evoked fMRI responses with behavior-based models that independently captured the visual and conceptual similarities among these stimuli. Our results revealed evidence for distinctive coding of visual features in lateral occipital cortex, and conceptual features in the temporal pole and parahippocampal cortex. By contrast, we found evidence for integrative coding of visual and conceptual object features in perirhinal cortex. The neuroanatomical specificity of this effect was highlighted by results from a searchlight analysis. Taken together, our findings suggest that perirhinal cortex uniquely supports the representation of fully specified object concepts through the integration of their visual and conceptual features.

DOI: https://doi.org/10.7554/eLife.31873.001

*For correspondence:
cmarti97@gmail.com (CBM);
barense@psych.utoronto.ca (MDB)

Competing interests: The authors declare that no competing interests exist.

## Introduction

Semantic memory imbues the world with meaning and shapes our understanding of the relationships among object concepts. Many neurocognitive models of semantic memory incorporate the notion that object concepts are represented in a feature-based manner (*Rosch and Mervis, 1975*; *Tyler and Moss, 2001*; *Rogers and McClelland, 2004*). On this view, our understanding of the concept 'hairdryer' is thought to reflect knowledge of observable perceptual properties (e.g. visual form) and abstract conceptual features (e.g. '*used to style hair*'). Importantly, there is not always a one-to-one correspondence between how something looks and what it is; a hairdryer and a comb are conceptually similar despite being visually distinct, whereas a hairdryer and a gun are conceptually distinct despite being visually similar. Thus, a fully-specified representation of an object concept (i.e. one that can be distinguished from any and all other concepts), requires integration of its perceptual and conceptual features.

Neuroimaging research suggests that object features are coded in the modality-specific cortical regions that supported their processing at the time of acquisition (*Thompson-Schill, 2003*). For example, knowledge about the visual form of an object concept is thought to be coded in occipito-temporal visual processing regions (*Martin and Chao, 2001*). However, neurocognitive models of semantic memory differ with respect to how distributed feature representations relate to fully specified object concepts. On one view, these representations are thought to emerge through interactions among modality-specific cortical areas (*Kiefer and Pulvermüller, 2012*; *Martin, 2016*).

**eLife digest** Our ability to interact with the world depends in large part on our understanding of objects. But objects that look similar, such as a hairdryer and a gun, may do different things, while objects that look different, such as tape and glue, may have similar roles. The fact that we can effortlessly distinguish between such objects suggests that the brain combines information about an object's visual and abstract properties.

Nevertheless, brain imaging experiments show that thinking about what an object looks like activates different brain regions to thinking about abstract knowledge. For example, thinking about an object's appearance activates areas that support vision, whereas thinking about how to use that object activates regions that control movement. So how does the brain combine these different kinds of information?

Martin et al. asked healthy volunteers to answer questions about objects while lying inside a brain scanner. Questions about appearance (such as "is a hairdryer angular?") activated different regions of the brain to questions about abstract knowledge ("is a hairdryer manmade?"). But both types of question also activated a region of the brain called the perirhinal cortex. When volunteers responded to either type of question, the activity in their perirhinal cortex signaled both the physical appearance of the object as well as its abstract properties, even though both types of information were not necessary for the task. This suggests that information in the perirhinal cortex reflects combinations of multiple features of objects.

These findings provide insights into a neurodegenerative disorder called semantic dementia. Patients with semantic dementia lose their general knowledge about the world. This leads to difficulties interacting with everyday objects. Patients may try to use a fork to comb their hair, for example. Notably, the perirhinal cortex is a brain region that is usually damaged in semantic dementia. Loss of combined information about the visual and abstract properties of objects may lie at the core of the observed impairments.

DOI: https://doi.org/10.7554/eLife.31873.002

Within a competing class of theories, they are thought to reflect the integration of modality-specific features in trans-modal convergence zones (*Damasio, 1989*; *Rogers et al., 2004*; *Binder and Desai, 2011*), such as the anterior temporal lobes (ATL) (*Patterson et al., 2007*; *Tranel, 2009*; *Ralph et al., 2017*).

The dominant view of the ATL as a semantic hub was initially shaped by neuropsychological investigations in individuals with semantic dementia (SD) (*Patterson et al., 2007*). Behaviorally, SD is characterized by the progressive loss of conceptual knowledge across all receptive and expressive modalities (*Warrington, 1975*; *Hodges et al., 1992*). At the level of neuropathology, SD is associated with extensive atrophy of the ATL, with the earliest and most pronounced volume loss in the left temporal pole (*Mummery et al., 2000*; *Galton et al., 2001*). Most important from a theoretical perspective, patients with SD tend to confuse conceptually similar objects that are visually distinct (e.g. hairdryer – comb), but not visually similar objects that are conceptually distinct (e.g., hairdryer – gun), indicating that the temporal pole expresses conceptual similarity structure (*Graham et al., 1994*; see *Peelen and Caramazza, 2012*; *Chadwick et al., 2016*, for related neuroimaging evidence). Taken together, these findings suggest that the temporal pole supports multi-modal integration of abstract conceptual, but not perceptual, features. Notably, however, a considerable body of research indicates that the temporal pole may not be the only ATL structure that supports feature-based integration.

The representational-hierarchical model of object coding emphasizes a role for perirhinal cortex (PRC), located in the medial ATL, in feature integration that is distinct from that of the temporal pole (*Murray and Bussey, 1999*). Namely, within this framework PRC is thought to support the integration of conceptual *and* perceptual features. In line with this view, object representations in PRC have been described in terms of conceptual feature conjunctions in studies of semantic memory (*Moss et al., 2005*; *Bruffaerts et al., 2013*; *Clarke and Tyler, 2014*; *2015*; *Wright et al., 2015*), and visual feature conjunctions in studies of visual processing (*Barense et al., 2005*; *2007*; *2012*; *Lee et al., 2005*; *Devlin and Price, 2007*; *Murray et al., 2007*; *O'Neil et al., 2009*;

*Graham et al., 2010*). However, it is difficult to synthesize results from these parallel lines of research, in part, because conceptual and perceptual features tend to vary concomitantly across stimuli (*Mur, 2014*). For example, demonstrating that 'horse' and 'donkey' are represented with greater neural pattern similarity in PRC than are 'horse' and 'dolphin' may reflect differences in conceptual or perceptual relatedness. Thus, although the representational-hierarchical account was initially formalized nearly two decades ago (*Murray and Bussey, 1999*), direct evidence of integration across conceptual and perceptual features remains elusive.

In the current study, we used fMRI to identify where in the brain visual and conceptual object features are stored, and to determine whether and where they are integrated at the level of fully specified object representations. To this end, we first generated behavior-based models that captured the visual and conceptual similarities among a set of object concepts, ensuring that these dimensions were not confounded across stimuli (*Figure 1*). Next, participants were scanned using task contexts that biased attention to either the conceptual or visual features of these well-characterized object concepts (*Figure 2*). We then used representational similarity analysis (RSA) (*Kriegeskorte and Kievit, 2013*), implemented using ROI- and searchlight-based approaches, to determine where the brain-based similarity structure among object-evoked multi-voxel activity patterns could be predicted by the similarity structure in the behavior-based visual and conceptual similarity models.

We predicted that lateral occipital cortex (LOC), an occipito-temporal region that has been implicated in the processing of visual form (*Grill-Spector et al., 1999*; *Kourtzi and Kanwisher, 2001*; *Milner and Goodale, 2006*), would represent stored visual object features in a visual similarity code. Based on the neurocognitive models of semantic memory reviewed, we predicted that the temporal pole would represent stored conceptual object features in a conceptual similarity code (*Patterson et al., 2007*; *Ralph et al., 2017*). We also predicted conceptual similarity coding in parahippocampal cortex, which has been linked to the representation of the contextually-based co-occurrence of objects (*Bar, 2004*; *Aminoff et al., 2013*). Critically, objects that are regularly encountered in the same context (e.g. 'comb' and 'hairdryer' in a barbershop) often share many conceptual features (e.g. '*used to style hair*'). Thus, to the extent that shared conceptual features directly shape contextual meaning, object-evoked responses in parahippocampal cortex may express conceptual similarity structure. Returning to the primary objective of the study, we predicted that PRC would uniquely represent the visual and conceptual features that define fully-specified object concepts in an integrated similarity code.

## Results

### Behavior-based similarity models

Using a data-driven approach, we first generated behavior-based models that captured the visual and conceptual similarities among 40 targeted object concepts (*Figure 1*). Notably, our visual similarity model and conceptual similarity model were derived from behavioral judgments provided by two independent groups of participants. For the purpose of constructing the visual similarity model, the first group of participants (N = 1185) provided pairwise comparative similarity judgments between object concepts (*Figure 1A*). Specifically, a pair of words was presented on each trial and participants were asked to rate the visual similarity between the object concepts to which they referred using a 5-point Likert scale. Similarity ratings for each pair of object concepts were averaged across participants, normalized, and expressed within a representational dissimilarity matrix (RDM). We refer to this RDM as the *behavior-based visual RDM*.

For the purpose of constructing the conceptual similarity model, a second group of participants (N = 1600) completed an online feature-generation task (*McRae et al., 2005*; *Taylor et al., 2012*) (*Figure 1B*). Each participant was asked to generate a list of conceptual features that characterize one object concept (e.g. hairdryer: '*used to style hair*', '*found in salons*', '*electrically powered*', '*blows hot air*'; comb: '*used to style hair*', '*found in salons*', '*has teeth*', '*made of plastic*'). Conceptual similarity between all pairs of object concepts was quantified as the cosine angle between the corresponding pairs of feature vectors. With this approach, high cosine similarity between object concepts reflects high conceptual similarity. Cosine similarity values were then expressed within an RDM, which we refer to as the *behavior-based conceptual RDM*.

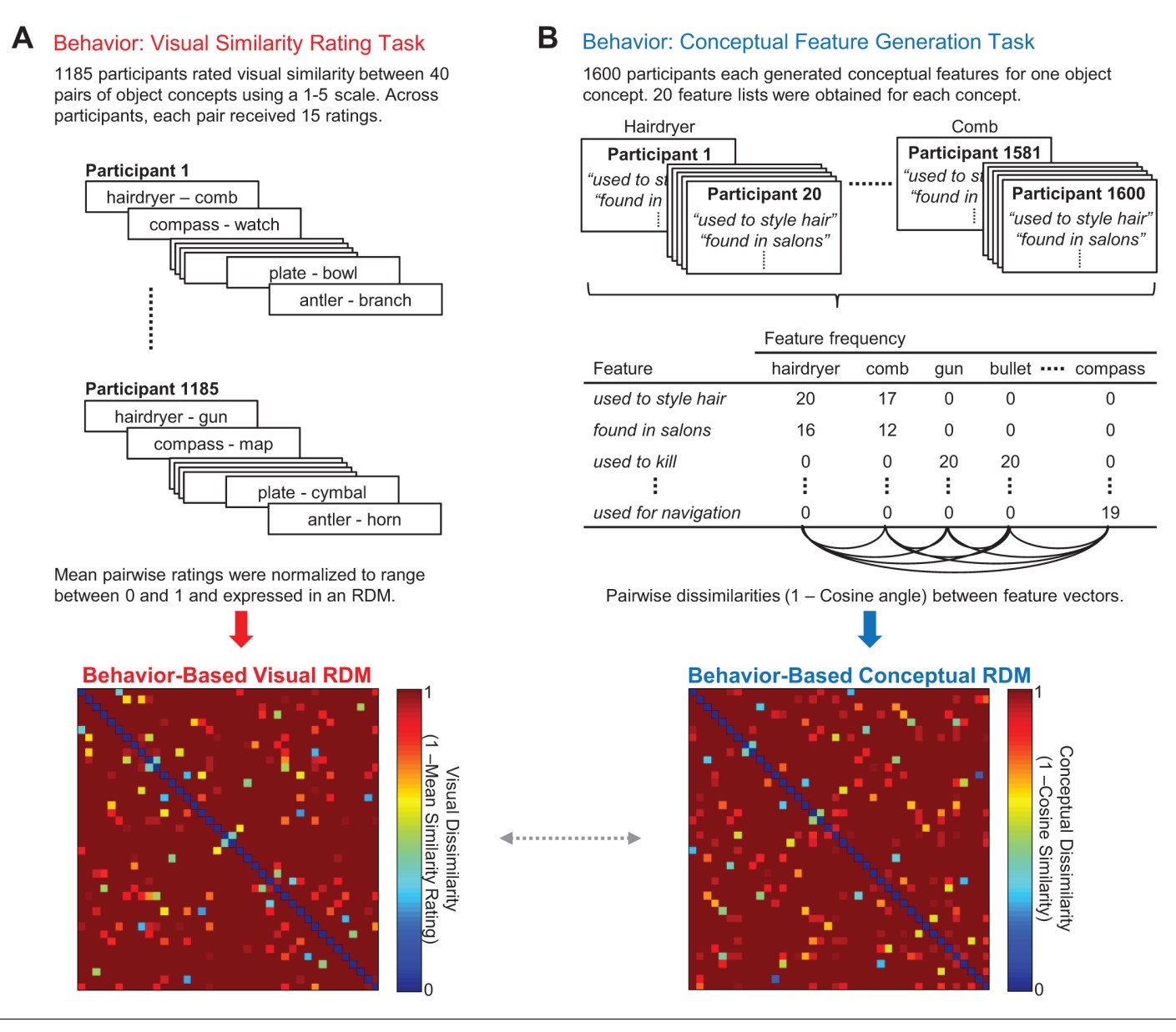

**Figure 1.** Behavior-based RDMs. (**A**) Visual similarity rating task (top) and corresponding 40 × 40 behavior-based visual RDM (bottom). (**B**) Conceptual feature generation task (top), abridged feature matrix depicting the feature frequencies across participants for each concept (middle), and corresponding 40 × 40 behavior-based conceptual RDM (bottom). The dashed horizontal arrow between behavior-based RDMs denotes a second-level RSA that compared these similarity models with one another. All object concepts are listed in *Figure 1—source data 1* - Object concepts and targeted pairs. Behavior-based RDMs (together with the word2vec RDM) are contained in *Figure 1—source data 2* - Behavior-based RDMs and word2vec RDM.
DOI: https://doi.org/10.7554/eLife.31873.003
The following source data is available for figure 1:

**Source data 1.** Object concepts and targeted pairs.
DOI: https://doi.org/10.7554/eLife.31873.004
**Source data 2.** Behavior-based RDMs and word2vec RDM.
DOI: https://doi.org/10.7554/eLife.31873.005

We next performed a second-level RSA to quantify the relationship between our behavior-based visual RDM and behavior-based conceptual RDM. This comparison is denoted by the gray arrow between behavior-based RDMs in *Figure 1*. Critically, this analysis revealed that the model RDMs were not significantly correlated with one another (Kendall's tau-a = 0.01, p=0.10), indicating that

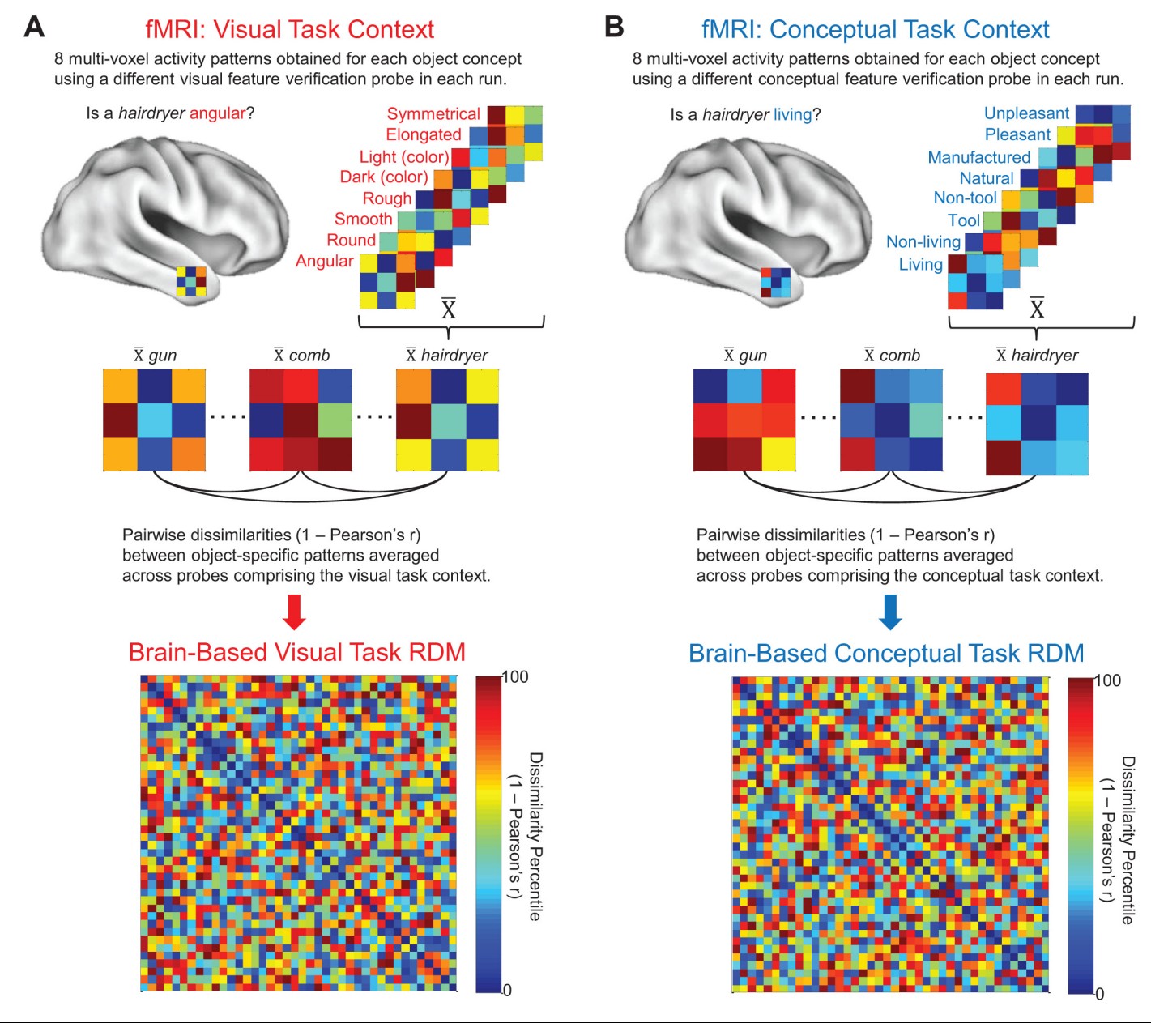

**Figure 2.** Brain-based RDMs. (A) Example of object-evoked neural activity patterns obtained across all eight probes in the visual task context (top), mean object-specific activity patterns averaged across repetitions (middle), and corresponding 40 × 40 brain-based visual task RDM derived from a first-level RSA (bottom). (B) Example of object-evoked neural activity patterns obtained across all eight probes in the conceptual task context (top), mean object-specific activity patterns averaged across repetitions (middle), and corresponding 40 × 40 brain-based conceptual task RDM derived from a first-level RSA (bottom).

DOI: https://doi.org/10.7554/eLife.31873.006

differences in visual and conceptual features were not confounded across object concepts. In other words, ensuring that these different types of features varied independently across stimuli (e.g. hairdryer – gun; hairdryer – comb), rather than concomitantly (e.g. horse – donkey; horse – dolphin), allowed us to isolate the separate influence of visual and conceptual features on the representational structure of object concepts in the brain. In this example, a hairdryer and a gun are visually similar but conceptually dissimilar, whereas a hairdryer and a comb are visually dissimilar but conceptually similar.

## Comparison of behavior-based RDMs with a corpus-based (word2vec) semantic RDM

We next sought to compare our behavior-based RDMs with a corpus-based model of conceptual similarity. To this end, we implemented a word2vec language model, which mapped 3 million words to 300 feature vectors in a high-dimensional space (*Mikolov et al., 2013*). The model was trained using ~100 billion words from a Google News dataset. From this model, we calculated the cosine similarity between feature vectors for all pairs of words in our stimulus set. These data were expressed in a 40 × 40 word2vec RDM (*Figure 1—source data 1* contains the word2vec RDM). Importantly, the word2vec RDM was significantly correlated with our behavior-based conceptual RDM (Kendall's tau-a = 0.11, SE = 0.0141, p<0.00001), suggesting that both models captured the conceptual similarity structure among the object concepts. However, the word2vec RDM was also significantly correlated with our behavior-based visual RDM (Kendall's tau-a = 0.04, SE = 0.0130, p<0.001). This result suggests that, in line with our objectives, the behavior-based conceptual RDM captured semantic similarity selectively defined as conceptual object features, whereas the word2vec RDM may have captured a broader definition of semantic similarity, that is, one that includes both visual semantics and abstract conceptual features. Consistent with this view, gun and hairdryer were conceptually unrelated in our behavior-based conceptual RDM (cosine = 0), whereas the word2vec RDM suggested modest conceptual similarity (cosine = 0.16). Although this difference is likely determined by multiple factors, it is important to note that gun and hairdryer had a relatively high visual similarity index in our behaviour-based visual RDM (normalized mean rating = 0.58). These data highlight a theoretically important distinction between our behaviorally derived conceptual feature-based statistics and corpus-based estimates of semantic similarity. Specifically, the former allow for distinctions between visual and conceptual object features, whereas corpus-based models may not.

## fMRI task and behavioral results

We used fMRI to estimate the representational structure of our 40 object concepts from neural activity patterns in an independent group of participants (*Figure 2*). Given our specific interest in understanding pre-existing representations of object concepts rather than bottom-up perceptual processing, all stimuli were presented as words. This approach ensured that conceptual and visual features were extracted from pre-existing representations of object concepts. That is to say, both conceptual and visual features were arbitrarily related to the physical input (i.e. the orthography of the word). By contrast, when pictures are used as stimuli, visual features are accessible from the pictorial cue, whereas conceptual features require abstraction from the cue. Functional brain data were acquired over eight experimental runs, each of which consisted of two blocks of stimulus presentation. All 40 object concepts were presented sequentially within each block, for a total of 16 repetitions per concept. On each trial, participants were asked to make a 'yes/no' property verification judgment in relation to a block-specific verification probe. Half of the blocks were associated with verification probes that encouraged processing of visual features (e.g. 'is the object angular?"), and the other half were associated with verification probes that encouraged processing of conceptual features (e.g. 'is the object a tool?"). Each run consisted of one visual feature verification block and one conceptual feature verification block, with order counterbalanced across runs. With this experimental design, we were able to characterize neural responses to object concepts across two task contexts: a visual task context (*Figure 2A*) and a conceptual task context (*Figure 2B*).

Behavioral performance on the scanned property verification task indicated that participants interpreted the object concepts and property verification probes with a high degree of consistency (*Figure 3*). Specifically, all participants (i.e. 16/16) provided the same yes/no response to the property verification task on 88.4% of all trials. Agreement was highest for the 'living' verification probe (96.8%) and lowest for the 'non-tool' verification probe (73.2%). Moreover, the proportion of trials on which all participants provided the same response did not differ between the visual feature verification task context (mean = 87.3% collapsed across all eight visual probes) and the conceptual feature verification task context (mean = 89.5% collapsed across all eight conceptual probes) (z = 0.19, p=0.85). Response latencies were also comparable across the visual feature verification task context (mean = 1361 ms, SD = 303) and the conceptual feature verification task context (mean = 1376 ms, SD = 315) (t (15)=1.00, p=0.33, 95% CI [−49.09, 17.71].

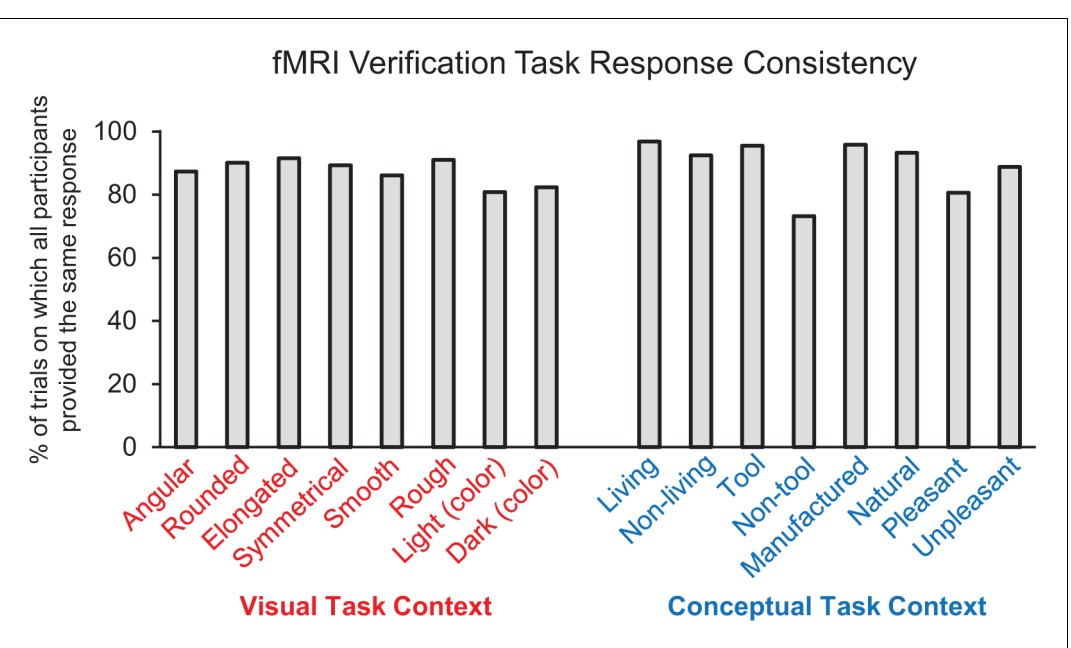

**Figure 3.** fMRI feature verification task performance. Percentage of trials on which all participants (i.e. 16/16) provided the same 'yes/no' response for each property verification probe.
DOI: https://doi.org/10.7554/eLife.31873.007

## ROI-based RSA: comparison of behavior-based RDMs with brain-based RDMs

We next quantified pairwise similarities between object-evoked multi-voxel activity patterns using a first-level RSA (*Figure 2*). For the purpose of conducting ROI-based RSA, we focused on multi-voxel activity patterns obtained in PRC, the temporal pole, parahippocampal cortex, and LOC. ROIs from a representative participant are presented in *Figure 4*. These ROIs were selected a priori based on empirical evidence linking their respective functional characteristics to visual object processing, conceptual object processing, or both. Our primary focus was on PRC, which has been linked to integrative coding of visual object features and conceptual object features across parallel lines of research (*Barense et al., 2005*; *2007*; *2012*; *Lee et al., 2005*; *O'Neil et al., 2009*; *Bruffaerts et al., 2013*; *Clarke and Tyler, 2014*; *2015*; *Wright et al., 2015*; *Erez et al., 2016*). The temporal pole has primarily been linked to processing of conceptual object properties (*Mummery et al., 2000*; *Galton et al., 2001*; *Patterson et al., 2007*; *Pobric et al., 2007*; *Lambon Ralph et al., 2009*; *Peelen and Caramazza, 2012*; *Chadwick et al., 2016*). Parahippocampal cortex has been implicated in the conceptual processing of contextual associations, including representing the co-occurrence of objects, although its functional contributions remain less well defined than the temporal pole (*Bar and Aminoff, 2003*; *Aminoff et al., 2013*; *Ranganath and Ritchey, 2012*). Lastly, LOC, which is a functionally defined region in occipito-temporal cortex, has been revealed to play a critical role in processing visual form (*Grill-Spector et al., 1999*; *Kourtzi and Kanwisher, 2001*; *Milner and Goodale, 2006*). Because we did not have any a priori predictions regarding hemispheric differences, estimates of neural pattern similarities between object concepts were derived from multi-voxel activity collapsed across ROIs in the left and right hemisphere.

Object-specific multi-voxel activity patterns were estimated in each run using general linear models fit to data from the visual and conceptual task contexts, separately. Mean object-specific responses were then calculated for each task context by averaging across runs. Linear correlation distances (Pearson's r) were calculated between all pairs of object-specific multi-voxel activity patterns within each task context and expressed in participant-specific *brain-based visual task RDMs* and *brain-based conceptual task RDMs*. The brain-based visual task RDMs captured the neural pattern similarities obtained between all object concepts in the visual task context (i.e. while participants made visual feature verification judgments) (*Figure 2A*), and the brain-based conceptual task

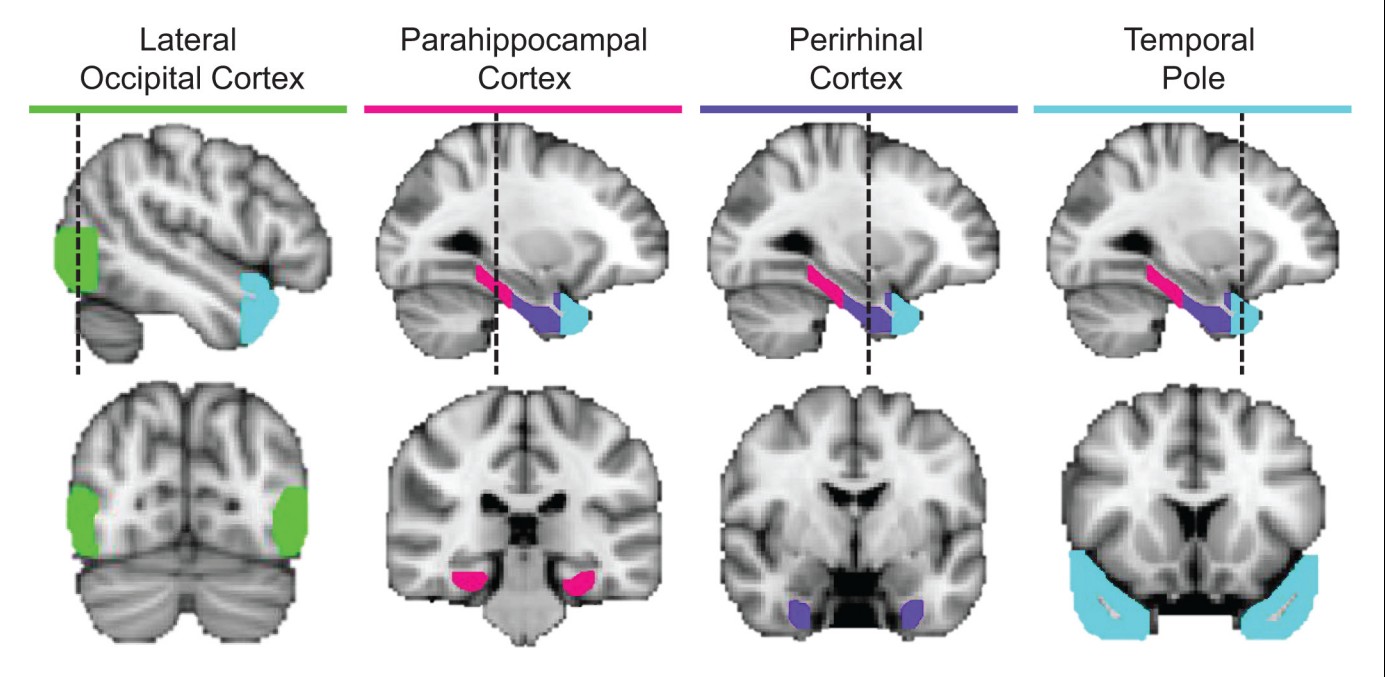

**Figure 4.** ROIs in a representative participant. Cortical regions examined in the ROI-based RSAs, including lateral occipital cortex (green), parahippocampal cortex (pink), perirhinal cortex (purple), and the temporal pole (cyan).
DOI: https://doi.org/10.7554/eLife.31873.008

RDMs captured the neural pattern similarities obtained between all object concepts in the conceptual task context (i.e. while participants made conceptual feature verification judgments) (*Figure 2B*).

We implemented second-level RSA to compare behavior-based visual and conceptual RDMs with the brain-based visual and conceptual task RDMs (these comparisons are denoted by the solid vertical and diagonal arrows in *Figure 5*). All RDMs were compared in each ROI using a ranked correlation coefficient (Kendall's tau-a) as a similarity index (*Nili et al., 2014*). Inferential statistical analyses were performed using a one-sided Wilcoxon signed-rank test, with participants as a random factor. A Bonferroni correction was applied to adjust for multiple comparisons (4 ROIs x 2 behavior-based RDMs x 2 brain-based RDMs = 16 comparisons, yielding a critical alpha of. 003). With this approach, we revealed that object concepts are represented in three distinct similarity codes that differed across ROIs: a visual similarity code, a conceptual similarity code, and an integrative code. Results from our ROI-based RSA analyses are shown in *Figure 6* and discussed in turn below.

### Lateral occipital cortex represents object concepts in a task-dependent visual similarity code

Consistent with its well-established role in the processing of visual form, patterns of activity within LOC reflected the visual similarity of the object concepts (*Figure 6A*). Specifically, the brain-based visual task RDMs obtained across participants in LOC were significantly correlated with the behavior-based visual RDM (Kendall's tau-a = 0.045, p<0.002), but not the behavior-based conceptual RDM (Kendall's tau-a = −0.006, p=0.72). In other words, activity patterns in LOC expressed a visual similarity structure when participants were asked to make explicit judgments about the visual features that characterized object concepts (e.g. whether an object is angular in form). By contrast, the brain-based conceptual task RDMs obtained across participants in LOC were not significantly correlated with either the behavior-based visual RDM (Kendall's tau-a = 0.006, p=0.13) or the behavior-based conceptual RDM (Kendall's tau-a = 0.003, p=0.65). That is to say, activity patterns in LOC expressed neither visual nor conceptual similarity structure when participants made judgments that pertained to conceptual object features (e.g. whether an object is naturally occurring). Considered together, these results suggest that LOC represented perceptual information about object concepts in a task-

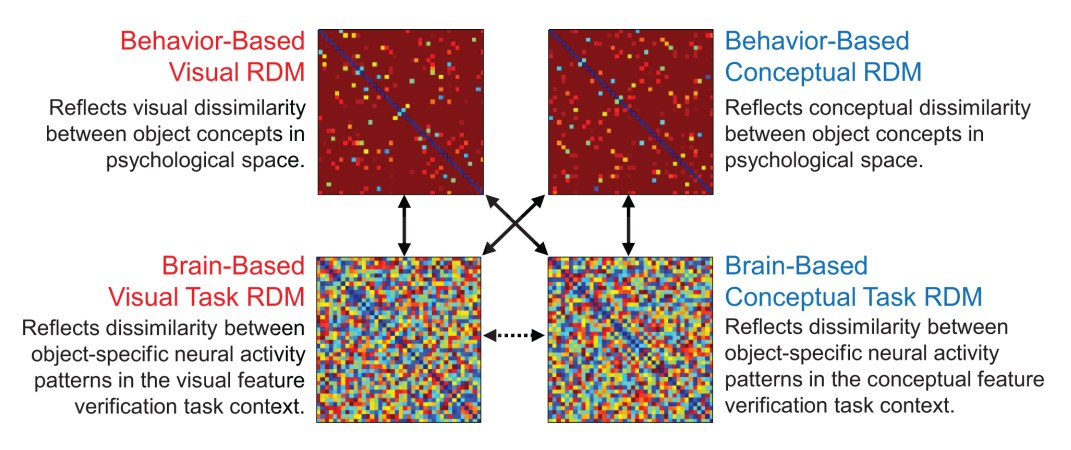

**Figure 5.** Second-level RSAs. Solid vertical and diagonal arrows reflect second-level RSA in which behavior-based RDMs were compared with brain-based RDMs (ROI-based results in *Figure 6*, searchlight-based results in *Figures 9*, *10* and *11*). The dashed horizontal arrow between brain-based RDMs reflects second-level RSA in which neural pattern similarities from each task context were directly compared with each other (results in *Figure 7*).
DOI: https://doi.org/10.7554/eLife.31873.009

dependent visual similarity code. Specifically, when task demands biased attention toward visual features, signals in LOC generalized across visually related object concepts even when they are conceptually distinct (e.g. hairdryer – gun).

## Parahippocampal cortex represents object concepts in a task-dependent conceptual similarity code

Patterns of activity obtained in parahippocampal cortex, which has previously been associated with the processing of semantically-based contextual associations (*Bar and Aminoff, 2003*; *Aminoff et al., 2013*), reflected the conceptual similarity of the object concepts (*Figure 6B*). First, the brain-based visual task RDMs obtained across participants in parahippocampal cortex were not significantly correlated with either the behavior-based visual RDM (Kendall's tau-a = 0.005, p=0.26) or the behavior-based conceptual RDM (Kendall's tau-a = 0.009, p=0.26). In other words, activity patterns in parahippocampal cortex expressed neither visual nor conceptual similarity structure when participants made judgments that pertained to conceptual object features (e.g. whether an object is symmetrical). Second, the brain-based conceptual task RDMs obtained across participants in parahippocampal cortex were not significantly related to the behavior-based visual RDM (Kendall's tau-a = −0.008, p=0.55), but they were correlated with the behavior-based conceptual RDM (Kendall's tau-a = 0.046, p<0.002). Thus, activity patterns in parahippocampal cortex expressed a conceptual similarity structure when participants were asked to make explicit judgments about the conceptual features that characterized object concepts (e.g. whether an object is a tool). Put another way, conceptual information was represented in parahippocampal cortex in a task-dependent manner that generalized across conceptually related object concepts even when they were visually distinct (e.g. hairdryer – comb).

## The temporal pole represents object concepts in a task-invariant conceptual similarity code

In line with theoretical frameworks that have characterized the temporal pole as a semantic hub (*Patterson et al., 2007*; *Tranel, 2009*), patterns of activity within this specific ATL structure reflected the conceptual similarity of the object concepts (*Figure 6D*). Specifically, whereas the brain-based visual task RDMs obtained across participants in the temporal pole were not significantly correlated with the behavior-based visual RDM (Kendall's tau-a = 0.006, p=0.25), they were correlated with the behavior-based conceptual RDM (Kendall's tau-a = 0.035, p<0.001). In other words, activity patterns in the temporal pole expressed a conceptual similarity structure when participants were asked to make explicit judgments about the visual features that characterized object concepts (e.g. whether

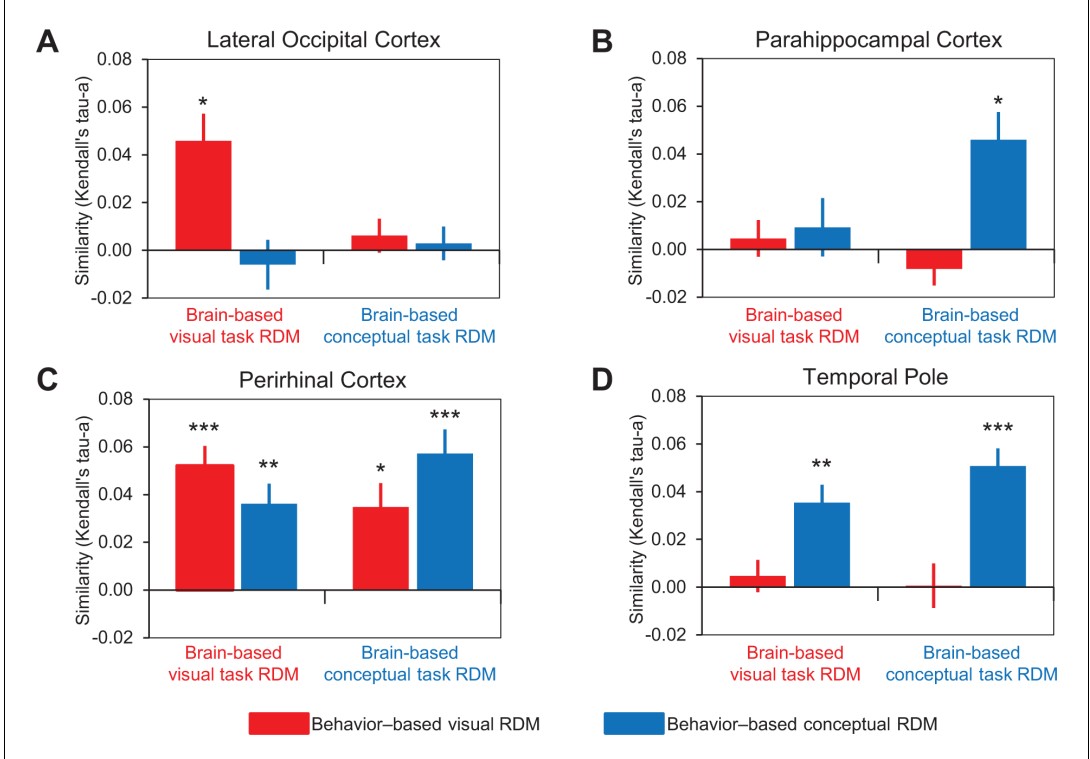

**Figure 6.** Comparison of behavior-based and brain-based RDMs. Similarities between behavior-based and brain-based RDMs are plotted for (A) LOC, (B) parahippocampal cortex, (C) PRC, and (D) the temporal pole. These comparisons are denoted by the solid vertical and diagonal arrows in *Figure 5*. Similarity was quantified as the ranked correlation coefficient (Kendall's tau-a) between behavior-based RDMs and the brain-based RDMs. Error bars indicate standard error of the mean. ***p<0.001, **p<0.01, *p<0.05 (Bonferroni corrected). Participant-specific Kendall's tau-a co-efficients are contained in *Figure 6—source data 1* - Comparison of similarity models and brain-based RDMs.

DOI: https://doi.org/10.7554/eLife.31873.010

The following source data and figure supplement are available for figure 6:

**Source data 1.** Comparison of similarity models and brain-based RDMs.
DOI: https://doi.org/10.7554/eLife.31873.012

**Figure supplement 1.** Comparison of word2vec RDM with brain-based RDMs.
DOI: https://doi.org/10.7554/eLife.31873.011

an object is elongated). Similarly, whereas the brain-based conceptual task RDMs obtained across participants in the temporal pole were not significantly correlated with the behavior-based visual RDM (Kendall's tau-a = 0.0005, p=0.47), they were correlated with the behavior-based conceptual RDM (Kendall's tau-a = 0.05, p<0.0001). Thus, activity patterns in the temporal pole expressed a conceptual similarity structure when participants were asked to make explicit judgments about *either* the visual or conceptual features that characterized object concepts (e.g. whether an object is dark in color, or whether an object is pleasant). In other words, conceptual information was represented in the temporal pole in a task-invariant manner that generalized across conceptually related object concepts even when they were visually distinct (e.g. hairdryer – comb).

### Perirhinal cortex represents object concepts in a task-invariant similarity code that reflects integration of visual and conceptual features

Results obtained in PRC support the notion that this structure integrates visual and conceptual object features (6C), as first theorized in the representational-hierarchical model of object representation (*Murray and Bussey, 1999*). Namely, we revealed that the brain-based visual task RDMs obtained across participants in PRC were significantly correlated with both the behavior-based visual RDM (Kendall's tau-a = 0.052, p<0.0001), and the behavior-based conceptual RDM (Kendall's tau-

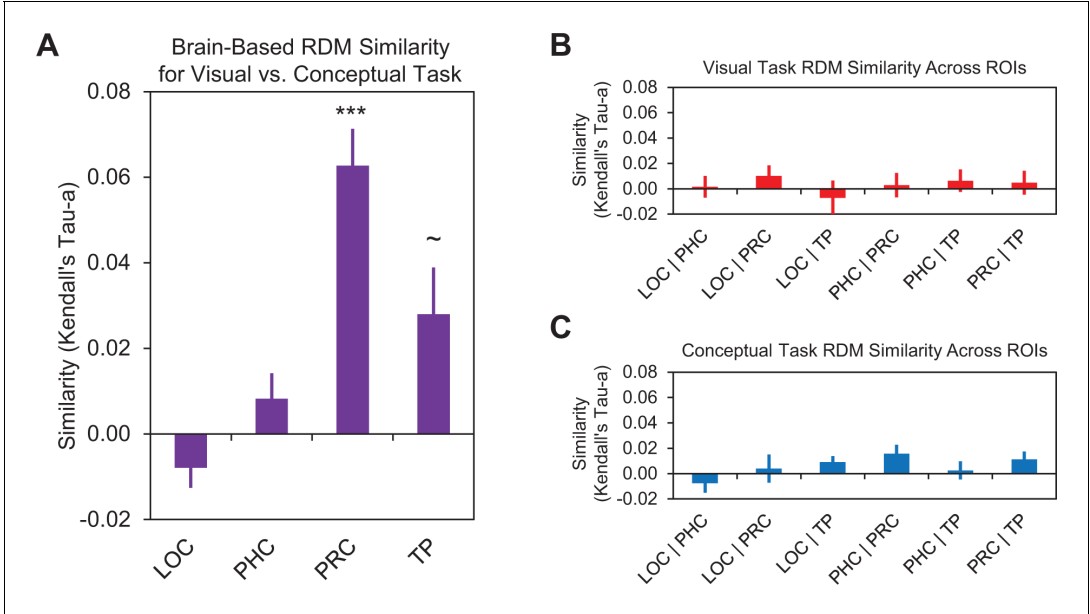

**Figure 7.** Comparison of brain-based RDMs. (**A**) Similarities between brain-based visual task RDMs and brain-based conceptual task RDMs within lateral occipital cortex (LOC), parahippocampal cortex (PHC), perirhinal cortex (PRC), and the temporal pole (TP). These comparisons are denoted by the dashed horizontal arrow in the bottom of *Figure 5*. (**B**) Similarities between brain-based visual task RDMs across different ROIs. Labels on the x-axis denote the ROIs being compared. (**C**) Similarities between brain-based conceptual task RDMs across different ROIs. Similarity was quantified as the ranked correlation coefficient (Kendall's tau-a) between behavior-based RDMs and the brain-based RDMs. Error bars indicate standard error of the mean. ***$p<0.001$ (Bonferroni corrected),~$p < 0.05$ (uncorrected). Participant-specific Kendall's tau-a co-efficients are contained in *Figure 7—source data 1* - Comparison of brain-based RDMs.

DOI: https://doi.org/10.7554/eLife.31873.013

The following source data is available for figure 7:

**Source data 1.** Comparison of brain-based RDMs.

DOI: https://doi.org/10.7554/eLife.31873.014

a = 0.036, $p<0.0003$). Similarly, the brain-based conceptual task RDMs obtained across participants were also correlated with both the behavior-based visual RDM (Kendall's tau-a = 0.035, $p<0.002$), and the behavior-based conceptual RDM (Kendall's tau-a = 0.057, $p<0.0001$). In other words, activity patterns in PRC expressed both visual and conceptual similarity structure when participants were asked to make explicit judgments about the visual features that characterized object concepts (e.g. whether an object is round) and when participants were asked to make explicit judgments about the conceptual features that characterized object concepts (e.g. whether an object is manufactured).

Numerically, patterns of activity in PRC showed more similarity to the behavior-based visual RDM than to the behavior-based conceptual RDM in the visual task context, and vice versa in the conceptual task context. Therefore, we performed a 2 [behavior-based RDMs] x 2 [brain-based task RDMs] repeated measures ANOVA to formally test for an interaction between behavior-based model and fMRI task context. For this purpose, all Kendall's tau-a values were transformed to Pearson's *r* co-efficients ($r = sin$ (½ π tau-a), *Walker, 2003*), which were then Fisher-z transformed. The task x model interaction neared, but did not reach, significance ($F(1,15) = 3.48$, $p=0.082$).

In sum, these findings indicate that PRC simultaneously expressed both conceptual and visual similarity structure, and did so regardless of whether participants were asked to make targeted assessments of conceptual or visual features. In other words, activity patterns in PRC captured the conceptual similarity between hairdryer and comb, as well as the visual similarity between hairdryer and gun, and did so irrespective of task context. Critically, these results were obtained despite the fact that the brain-based RDMs were orthogonal to one another (i.e. not significantly correlated). Considered together, these results suggest that, of the a priori ROIs considered, PRC represents object concepts at the highest level of specificity through integration of visual and conceptual features.

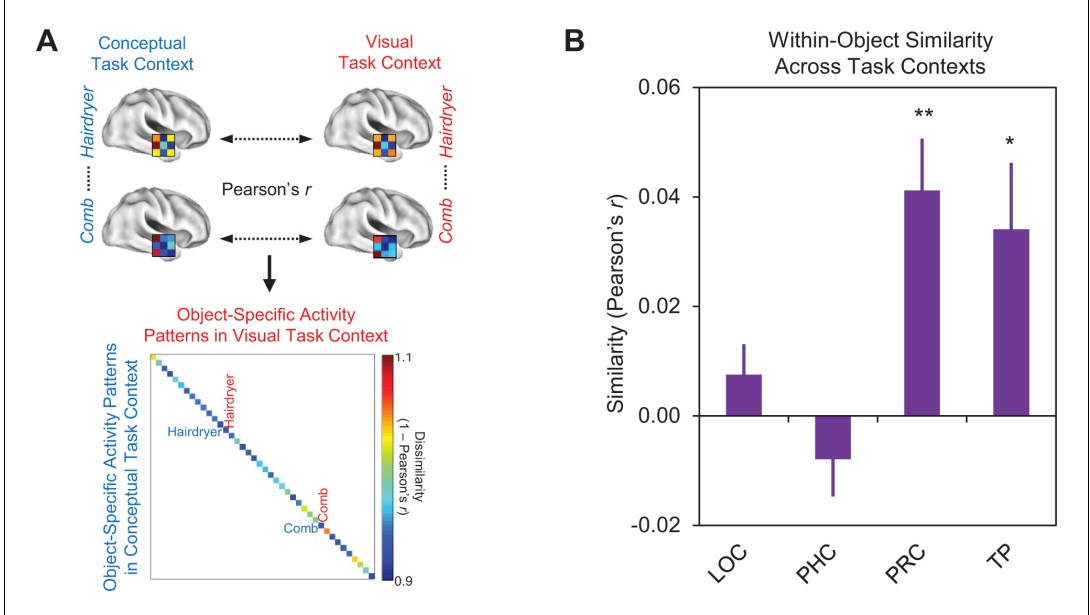

**Figure 8.** Comparison of within-object multi-voxel activity patterns across different task contexts. (A) Depiction of first-level RSA procedure for quantifying within-object multi-voxel activity patterns across the visual and conceptual task contexts. (B) Mean similarities between within-object multi-voxel activity patterns across different task contexts within each region of interest. Similarity was quantified as the linear correlation coefficient (Pearson's r) between object-evoked multi-voxel activity patterns. Lateral occipital cortex (LOC), parahippocampal cortex (PHC), perirhinal cortex (PRC), and the temporal pole (TP). Error bars indicate standard error of the mean. **$p<0.01$, *$p<0.05$ (Bonferroni corrected). Participant-specific Pearson's r coefficients are contained in *Figure 8—source data 1* - Comparison of within-object similarity across task contexts.
DOI: https://doi.org/10.7554/eLife.31873.015

The following source data is available for figure 8:

**Source data 1.** Comparison of within-object similarity across task contexts.
DOI: https://doi.org/10.7554/eLife.31873.016

## ROI-based RSA: comparison of corpus-based (word2vec) semantic RDM with brain-based RDMs

For the purpose of comparison, we next examined similarities between the word2vec RDM and the brain-based RDMs using the same procedures described in the previous section. Results are presented in *Figure 6—figure supplement 1*. These analyses revealed significant positive correlations between the word2vec RDM and the brain-based conceptual task RDMs in parahippocampal cortex (Kendall's tau-a = 0.05, $p<0.01$), PRC (Kendall's tau-a = 0.035, $p<0.01$), and the temporal pole (Kendall's tau-a = 0.029, $p<0.01$). The word2vec RDM was also significantly correlated with the brain-based visual task RDMs in PRC (Kendall's tau-a = 0.025, $p<0.05$) and the temporal pole (Kendall's tau-a = 0.027, $p<0.05$). Notably, this pattern of results was identical to that obtained using the behavior-based conceptual RDMs in parahippocampal cortex, PRC, and the temporal pole. Interestingly, however, the word2vec RDM was also significantly correlated with the brain-based visual task RDMs in LOC (Kendall's tau-a = 0.028, $p<0.05$). This result is consistent with the observation that the word2vec RDM was significantly correlated with our behavior-based visual RDM, and further suggests that corpus-based models of semantic memory likely capture similarities between object concepts at the level of abstract conceptual properties and visual semantics.

## ROI-based RSA: comparisons of brain-based RDMs within ROIs

Having examined the relationships between behavior-based RDMs and brain-based RDMs, we next sought to directly characterize the relationships between brain-based conceptual and visual RDMs within each ROI (these comparisons are denoted by the dashed horizontal arrow in the bottom of *Figure 5*). These analyses were conducted using the same methodological procedures used to compare behavior-based RDMs with brain-based RDMs in the previous section. A Bonferroni correction

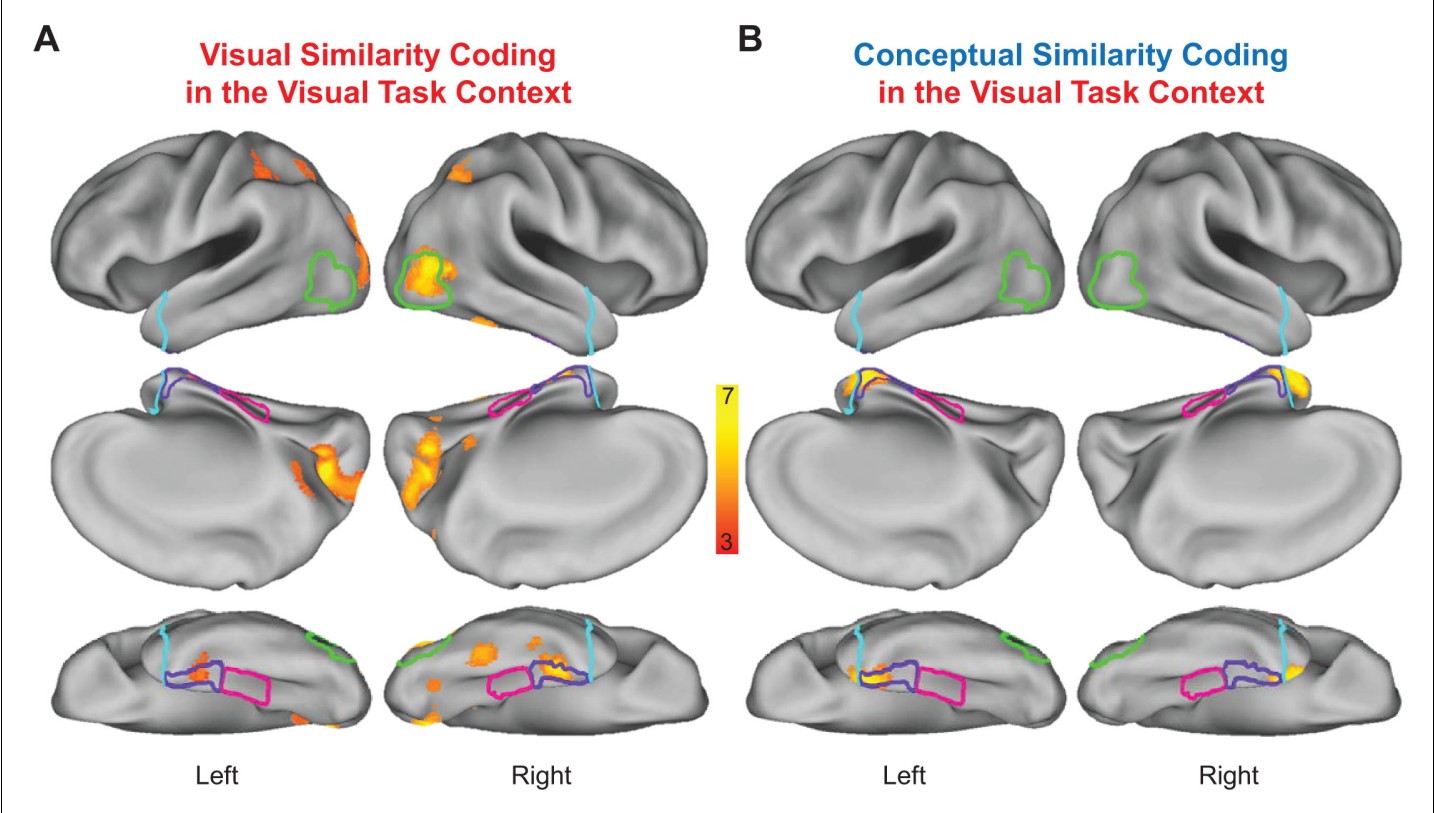

**Figure 9.** Visual task context representational similarity searchlight mapping results. (A) Cortical regions in which the brain-based visual task RDMs were significantly correlated with the behavior-based visual RDM. (B) Cortical regions in which the brain-based visual task RDMs were significantly correlated with the behavior-based conceptual RDM. The correlation coefficients (Kendall's tau-a) obtained between behavior-based RDMs and brain-based RDMs were Fisher-$z$ transformed and mapped to the voxel at the centre of each searchlight. Similarity maps were corrected for multiple comparisons using threshold-free cluster enhancement with a corrected statistical threshold of $p<0.05$ on the cluster level (*Smith and Nichols, 2009*). Outlines are shown for the lateral occipital cortex (green), parahippocampal cortex (pink), perirhinal cortex (purple), and the temporal pole (cyan).
DOI: https://doi.org/10.7554/eLife.31873.017

was applied to adjust for multiple comparisons (16 brain-based comparisons, yielding a critical alpha of. 003). Using second-level RSAs, we asked whether the brain-based visual task RDMs and brain-based conceptual task RDMs had a common similarity structure within a given ROI. Results are plotted in *Figure 7A*. Importantly, we found a significant positive correlation in PRC (Kendall's tau-a = 0.063, p<0.0001), and a trend toward a significant correlation in the temporal pole (Kendall's tau-a = 0.032, p=0.012). Conversely, brain-based visual and conceptual task RDMs were not significantly correlated in either parahippocampal cortex (Kendall's tau-a = 0.008, p=0.12), or LOC (Kendall's tau-a = −0.008, p=0.92). These results suggest that object concepts were represented similarly within PRC, and to a lesser extent within the temporal pole, regardless of whether they were encountered in a visual or conceptual task context.

### ROI-based RSA: comparisons of brain-based RDMs across ROIs

We next conducted second-level RSAs to quantify representational similarities between the brain-based visual task RDMs obtained across different ROIs. In other words, we asked whether activity in different ROIs (e.g. PRC and LOC) reflected similar representational distinctions across object concepts within the visual task context. Results are plotted in *Figure 7B*. Interestingly, these analyses did not reveal any significant results between any of our ROIs (all Kendall's tau-a <0.01, all p>0.07). These findings indicate that PRC and LOC, two regions that expressed a visual similarity code, represented different aspects of the visual object features.

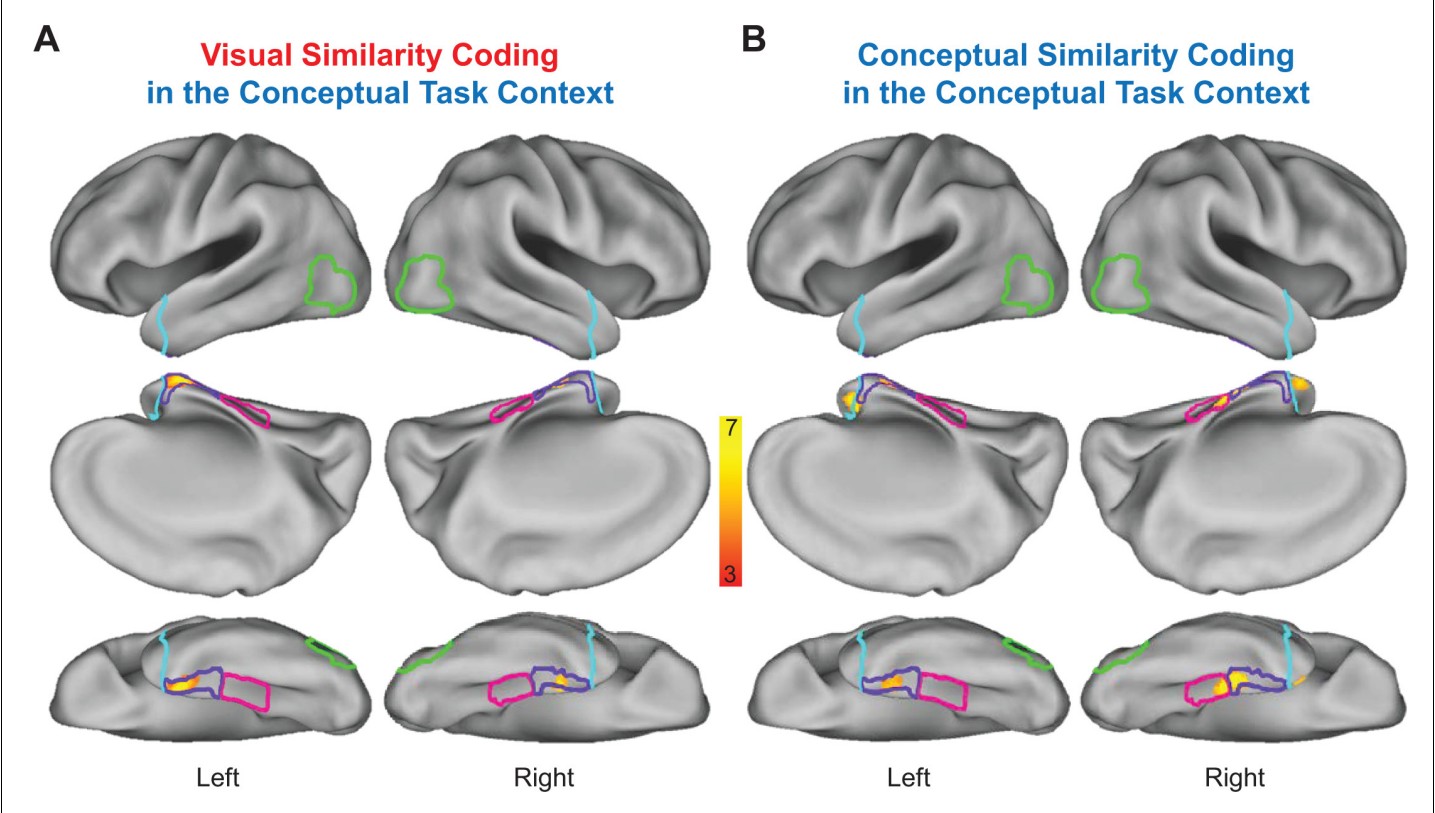

**Figure 10.** Conceptual task context representational similarity searchlight mapping results. (**A**) Cortical regions in which the brain-based conceptual task RDMs were significantly correlated with the behavior-based visual RDM. (**B**) Cortical regions in which the brain-based conceptual task RDMs were significantly correlated with the behavior-based conceptual RDM. The correlation coefficients (Kendall's tau-a) obtained between behavior-based RDMs and brain-based RDMs were Fisher-*z* transformed and mapped to the voxel at the centre of each searchlight. Similarity maps were corrected for multiple comparisons using threshold-free cluster enhancement with a corrected statistical threshold of p<0.05 on the cluster level (*Smith and Nichols, 2009*). Outlines are shown for the lateral occipital cortex (green), parahippocampal cortex (pink), perirhinal cortex (purple), and the temporal pole (cyan).

DOI: https://doi.org/10.7554/eLife.31873.018

Finally, we quantified the representational similarities between the brain-based conceptual task RDMs obtained across different ROIs. In other words, we asked whether activity in different ROIs (e.g. PRC and the temporal pole) reflected similar representational distinctions across object concepts within the conceptual task context. Results are plotted in *Figure 7C*. This set of analyses did not reveal any significant results between any of our ROIs (all Kendall's tau-a <0.016, all p>0.012). These findings indicate that the three regions that expressed a conceptual similarity code (i.e., PRC, parahippocampal cortex, and temporal pole), represented different aspects of the conceptual object features.

### ROI-based RSA: comparisons of within-object multi-voxel activity patterns across different task contexts

The RSAs reported thus far have quantified relationships among behavior-based and brain-based RDMs that reflected similarities between different object concepts (e.g. between 'hairdryer' and 'comb'). We next quantified within-object similarities (e.g. between 'hairdryer' and 'hairdryer') across visual and conceptual task contexts (e.g. 'is it living?' or 'is it angular?') using first-level RSAs. Specifically, we calculated one dissimilarity value (1 – Pearson's *r*) between the mean multi-voxel activity patterns evoked by a given object concept across different task contexts. These 40 within-object dissimilarity values were expressed along the diagonal of an RDM for each ROI in each participant,

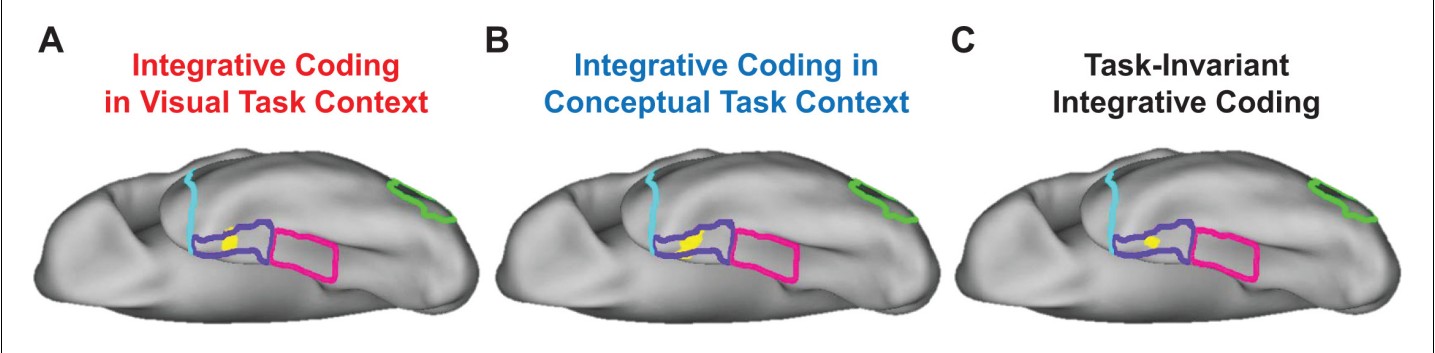

**Figure 11.** Overlap of searchlight similarity maps. (A) Overlap between similarity maps obtained in the visual task context (i.e. overlapping voxels from *Figure 9A and B*). (B) Overlap between similarity maps obtained in the conceptual task context (i.e. overlapping voxels from *Figure 10A and B*). (C) Overlap across brain-behavior similarity maps across both task contexts (i.e. overlapping voxels from *Figures 9A, B, 10A and B*). Outlines are shown for the lateral occipital cortex (green), parahippocampal cortex (pink), perirhinal cortex (purple), and the temporal pole (cyan).

DOI: https://doi.org/10.7554/eLife.31873.020

separately (*Figure 8A*). We next calculated mean within-object dissimilarity by averaging across the diagonal of each RDM for the purpose of performing statistical inference.

Results are presented in *Figure 8B*. Within-object similarity did not differ from zero in either LOC (Pearson's $r = 0.007$, p=0.20) or parahippocampal cortex (Pearson's $r = -0.008$, p=0.87), suggesting that a given object concept was represented differently across the visual and conceptual task contexts in these ROIs. These findings are consistent with the task-dependent nature of the similarity codes we observed in these regions (*Figure 6A and B*). Conversely, within-object similarity was significantly greater than zero in the temporal pole (Pearson's $r = 0.34$, p<0.05, Bonferroni corrected for four comparisons), indicating that this structure represents a given object concept similarly across different task contexts. This observation is consistent with results from the previous section which revealed that the similarities between object concepts in the temporal pole are preserved across task contexts (*Figure 7A*). These findings reflected the fact that the same conceptual object information (e.g. '*used to style hair*' and '*found in salons*') was carried in multi-voxel activity patterns obtained in each task context (*Figure 6D*). Within-object similarity was also significantly greater than zero in PRC (Pearson's $r = 0.41$, p<0.01, Bonferroni corrected for four comparisons), again indicating that a given object concept was represented similarly across different task contexts. This finding dovetails with our result from the previous section which revealed that the similarities between object concepts in PRC were preserved across task contexts (*Figure 7A*). When considered together, we interpret this pattern of results in PRC as further evidence of integrative coding, reflecting the fact that this structure carried the same conceptual (e.g. '*used to style hair*' and '*found in sal-ons*') and visual (e.g. visually similar to a gun) object information in both task contexts (*Figure 6C*).

## Searchlight-based RSA: comparisons of behavior-based RDMs with brain-based RDMs

### Perirhinal cortex is the only cortical region that supports integrative coding of conceptual and visual object features

We next implemented a whole-volume searchlight-based RSA to further characterize the neuroanatomical specificity of our ROI-based results. Specifically, we sought to determine whether object representations in PRC expressed visual and conceptual similarity structure within overlapping or distinct populations of voxels. If PRC does indeed support the integrative coding of visual and conceptual object features, then the same set of voxels should express both types of similarity codes. If PRC does not support the integrative coding of visual and conceptual object features, then different subsets of voxels should express these different similarity codes. More generally, data-driven searchlight mapping allowed us to explore whether any other regions of the brain showed evidence for integrative coding of visual and conceptual features in a manner comparable to that observed in PRC. To this end, we performed searchlight RSA using multi-voxel activity patterns restricted to a

100 voxel ROI that was iteratively swept across the entire cortical surface (*Kriegeskorte et al., 2006*; *Oosterhof et al., 2011*). In each searchlight ROI, the behavior-based RDMs were compared with the brain-based RDMs using a procedure identical to that implemented in our ROI-based RSA. These comparisons are depicted by the solid black vertical and diagonal arrows in *Figure 5*. The obtained similarity values (Pearson's *r*) were Fisher-*z* transformed and mapped to the center of each ROI for each participant separately. With this approach, we obtained participant-specific similarity maps for all comparisons, which were then standardized and subjected to a group-level statistical analysis. A threshold-free cluster enhancement (TFCE) method was used to correct for multiple comparisons with a cluster threshold of p<0.05 (*Smith and Nichols, 2009*).

Statistically thresholded group-level similarity maps depicting cortical regions in which behavior- and brain-based RDMs were significantly correlated are presented for the visual task context in *Figure 9*, and for the conceptual task context in *Figure 10*. Corresponding cluster statistics, co-ordinates, and neuroanatomical labels are reported in *Table 1*. Importantly, results from our whole-volume searchlight mapping analysis showed a high degree of consistency with our ROI-based results. First, we found evidence for visual similarity coding in the visual task context in aspects of right LOC (*Figure 9A*), as well as aspects of early visual cortex, posterior parietal cortex, and areas of medial and lateral ventral temporal cortex (*Figure 9A*). Next, we revealed conceptual similarity coding in the conceptual task context within a cluster of voxels that straddled the border between right parahippocampal cortex and PRC (*Figure 10B*). Although this cluster was only partially situated with parahippocampal cortex, it is interesting to note that its posterior extent did slightly encroach upon anterior aspects of the parahippocampal place area (PPA; functionally defined using a group-level GLM (scenes > objects); *Epstein and Kanwisher, 1998*), which has previously been linked to the representation of abstract conceptual information (*Aminoff et al., 2007*; *Baldassano et al., 2013*; *Marchette et al., 2015*). Moreover, we found evidence of conceptual similarity coding in bilateral aspects of the temporal pole in both task contexts (*Figures 9B* and *10B*), an observation that is consistent with results from multiple prior studies that have demonstrated conceptual similarity structure this aspect of the ATL (*Peelen and Caramazza, 2012*; *Chadwick et al., 2016*; cf *Fairhall and Caramazza, 2013*). Finally, and most importantly, results from the whole-brain searchlight revealed evidence for visual similarity coding and conceptual similarity coding in PRC in both task contexts (*Figures 9* and *10*). This result dovetails with findings from previous RSA-based fMRI research that has demonstrated conceptual similarity coding in PRC (*Bruffaerts et al., 2013*; *Clarke and Tyler, 2014*; cf *Fairhall and Caramazza, 2013*).

Although suggestive, neither the searchlight- nor the ROI-based RSA results presented thus far necessarily imply integrative coding in PRC. Indeed, it is possible that visual and conceptual object information was carried in spatially distinct sub-regions of this structure. To examine this issue, we first asked whether any voxels showed both visual and conceptual similarity coding in the visual task context (similarity maps in *Figure 9A and B*, respectively) using a voxel overlap analysis (*Figure 11A*). Importantly, we revealed a contiguous cluster of voxels that was unique to left PRC in which both behavior-based RDMs predicted the similarity structure in the brain-based visual task RDMs. This result indicated that a subset of voxels in PRC carried information about visual *and* conceptual object information even when task demands biased attention toward visual object features. We next asked whether any voxels showed both visual and conceptual similarity coding in the conceptual task context (similarity maps in *Figure —figure supplement 10A and B*, respectively) using a second voxel overlap analysis (*Figure 11B*). This analysis also revealed a contiguous cluster of voxels that was unique to left PRC in which both behavior-based RDMs predicted the similarity structure in the brain-based conceptual task RDMs. This finding indicated that a subset of voxels in PRC carried information about visual *and* conceptual object information when task demands biased attention toward conceptual object features.

In a final step using a third voxel overlap analysis, we examined whether any voxels showed both visual and conceptual similarity coding in both the visual and conceptual task contexts (*Figure 11C*). This analysis revealed a contiguous cluster of voxels in left PRC in which both behavior-based RDMs predicted the similarity structure captured by both brain-based RDMs. This result indicated that a subset of voxels that were unique to PRC carried information about visual *and* conceptual object information regardless of whether task demands biased attention toward visual *or* conceptual object features. Ultimately, this pattern of results suggests that not only does PRC carry both visual and conceptual object information, but it does so in the same subset of voxels.

**Table 1.** Clusters in which behavior-based RDMs were significantly correlated with brain-based RDMs as revealed using representational similarity searchlight analyses, with corresponding cluster extent, peak z-values, and MNI co-ordinates[1].

| Region | Cluster extent | Peak z-value | X | Y | Z |
|---|---|---|---|---|---|
| **Visual task context** | | | | | |
| *Behavior-Based Visual RDM – Brain-Based Visual Task RDM* | | | | | |
| Mid calcarine | 1660 | 5.79 | -2 | −74 | 12 |
| R lateral occipital cortex | 455 | 3.89 | 50 | −66 | 4 |
| R perirhinal cortex | 112 | 3.64 | 34 | −12 | −34 |
| L superior parietal lobule | 110 | 3.21 | −32 | −40 | 44 |
| L perirhinal cortex | 76 | 2.85 | −30 | −12 | −36 |
| R superior parietal lobule | 48 | 2.64 | 38 | −54 | 54 |
| R fusiform gyrus | 45 | 2.77 | 40 | −46 | −20 |
| R precuneus | 29 | 2.66 | 12 | −76 | 48 |
| R Inferior Temporal Gyrus | 9 | 2.52 | 44 | −22 | −28 |
| *Behavior-Based Conceptual RDM – Brain-Based Visual Task RDM* | | | | | |
| L Perirhinal Cortex | 368 | 3.96 | −24 | 2 | −38 |
| R Perirhinal Cortex | 232 | 3.26 | 22 | 2 | −36 |
| *Overlap* | | | | | |
| L Perirhinal Cortex | 22 | | −30 | -8 | −38 |
| **Conceptual task context** | | | | | |
| *Behavior-Based Conceptual RDM – Brain-Based Conceptual Task RDM* | | | | | |
| L Perirhinal Cortex | 79 | 2.88 | −30 | −10 | −34 |
| R Parahippocampal Cortex | 64 | 2.94 | 30 | −24 | −24 |
| L Temporal Pole | 61 | 2.89 | −34 | 4 | −26 |
| R Temporal Pole | 25 | 2.70 | 24 | 12 | −36 |
| *Behavior-Based Visual RDM – Brain-Based Conceptual Task RDM* | | | | | |
| L Perirhinal Cortex | 98 | 4.87 | −26 | -4 | −10 |
| R Perirhinal Cortex | 26 | 3.01 | 28 | −12 | −34 |
| *Overlap* | | | | | |
| L Perirhinal Cortex | 31 | | −26 | -8 | −42 |
| **Overlap across all Behavior-Based RDMs and Brain-Based RDMs** | | | | | |
| L Perirhinal Cortex | 16 | | −30 | -8 | −36 |

[1]MNI co-ordinates are reported for the peak voxel in individual clusters and the centre of mass for cluster overlap.

DOI: https://doi.org/10.7554/eLife.31873.019

## Discussion

Decades of research has been aimed at understanding how object concepts are represented in the brain (*Warrington, 1975*; *Hodges et al., 1992*; *Martin et al., 1995*; *Murray and Bussey, 1999*; *Chen et al., 2017*), yet the fundamental question of whether and where their visual and conceptual features are integrated remains unanswered. Progress toward this end has been hindered by the fact that these features tend to vary concomitantly across object concepts. Here, we used a data-driven approach to systematically select a set of object concepts in which visual and conceptual features varied independently (e.g. hairdryer – comb, which are conceptually similar but visually distinct; hairdryer – gun, which are visually similar but conceptually distinct). Using RSA of fMRI data, we revealed novel evidence of task-dependent visual similarity coding in LOC, task-dependent

conceptual similarity coding in parahippocampal cortex, task-invariant coding in the temporal pole, and task-invariant integrative coding in PRC.

Several aspects of our data provide novel support for the notion that PRC uniquely represents the visual and conceptual features that define fully specified object concepts in an integrated similarity code. First, this was the only region of the brain in which both visual *and* conceptual object coding was revealed. Moreover, these effects were observed regardless of whether fMRI task demands biased attention toward visual *or* conceptual object features. These results are particularly striking given the fact that they were revealed using a behavior-based visual similarity model and a behavior-based conceptual similarity model that were orthogonal to one another. In other words, the degree of similarity between multi-voxel activity patterns obtained while participants made conceptual judgments, such as whether a 'hairdryer' is man-made or a 'gun' is pleasant, was captured by the degree of visual similarity between these object concepts. Likewise, the degree of similarity between multi-voxel activity patterns obtained while participants made visual judgments, such as whether a 'hairdryer' is angular or a 'comb' is elongated, was captured by the degree of conceptual similarity between these object concepts. In both cases, PRC carried information about pre-existing representations of object features that were neither required to perform the immediate task at hand, nor correlated with the features that did in fact have task-relevant diagnostic value. Moreover, we also found that the brain-based visual task RDMs and brain-based conceptual task RDMs were correlated with one another across task contexts in PRC. That is to say, the similarity between 'hairdryer' and 'gun' was comparable regardless of whether task demands biased attention toward visual or conceptual features. Likewise, we also revealed that PRC also represented a given object concept similarly across task contexts, that is, 'hairdryer' evoked a pattern of activation that was comparable across task contexts. When considered together, these results suggest that, at the level of PRC, it may not be possible to fully disentangle conceptual and perceptual information. An important but challenging objective for future research will be to determine whether this pattern of results can be replicated at the level of individual neurons.

What is the behavioral relevance of fully specified object representations in which visual and conceptual features are integrated? It has previously been suggested that such representations allow for discrimination among stimuli with extensive feature overlap, such as exemplars from the same category (*Murray and Bussey, 1999*; *Noppeney et al., 2007*; *Graham et al., 2010*; *Clarke and Tyler, 2015*). In line with this view, individuals with medial ATL lesions that include PRC typically have more pronounced conceptual impairments related to living than non-living things (*Warrington and Shallice, 1984*; *Moss et al., 1997*; *Bozeat et al., 2003*), and more striking perceptual impairments for objects that are visually similar as compared to visually distinct (*Barense et al., 2007*, *Barense et al., 2010*; *Lee et al., 2006*). Functional MRI studies in neurologically healthy individuals have also demonstrated increased PRC engagement for living as compared to non-living objects (*Moss et al., 2005*), for known as compared to novel faces (*Barense et al., 2011*; *Peterson et al., 2012*), and for faces or conceptually meaningless stimuli with high feature overlap as compared to low feature overlap (*O'Neil et al., 2009*; *Barense et al., 2012*). In a related manner, fully specified object representations in PRC have also been implicated in long-term memory judgments. For example, PRC has been linked to explicit recognition memory judgments when previously studied and novel items are from the same stimulus category (*Martin et al., 2013*; *2016*; *2018*), and when subjects make judgments about their lifetime of experience with a given object concept (*Duke et al., 2017*). Common among these task demands is the requirement to discriminate among highly similar stimuli. In such scenarios, a fully specified representation that reflects the integration of perceptual and conceptual features necessarily enables more fine-grained distinctions than a purely perceptual or conceptual representation.

This study also has significant implications for prominent neurocognitive models of semantic memory in which the ATL is characterized as a semantic hub (*Rogers et al., 2006*; *Patterson et al., 2007*; *Tranel, 2009*). On this view, the bilateral ATLs are thought to constitute a trans-modal convergence zone that abstracts conceptual information from the co-occurrence of features otherwise represented in a distributed manner across modality-specific cortical nodes. Consistent with this idea, we have shown that a behavior-based conceptual similarity model predicted the similarity structure of neural activity patterns in the temporal pole, irrespective of task context. Specifically, neural activity patterns associated with conceptually similar object concepts that are visually distinct (e.g. hairdryer – comb) were more comparable than were conceptually dissimilar concepts that are visually

similar (e.g. hairdryer – gun), even when task demands required a critical assessment of visual features. This observation, together with results obtained in PRC, demonstrates a representational distinction between these ATL structures, a conclusion that dovetails with recent evidence indicating that this region is not functionally homogeneous (*Binney et al., 2010*; *Murphy et al., 2017*). Ultimately, this outcome suggests that some ATL sub-regions play a prominent role in task-invariant extraction of conceptual object properties (e.g. temporal pole), whereas others appear to make differential contributions to the task-invariant integration of perceptual and conceptual features (e.g. PRC) (*Ralph et al., 2017*; *Chen et al., 2017*).

Convergent evidence from studies of functional and structural connectivity in humans, non-human primates, and rodents have revealed that PRC is connected to the temporal pole, parahippocampal cortex, LOC, and nearly all other unimodal and polymodal sensory regions in neocortex (*Suzuki and Amaral, 1994*; *Burwell and Amaral, 1998*; *Kahn et al., 2008*; *McLelland et al., 2014*; *Suzuki and Naya, 2014*; *Wang et al., 2016*; *Zhuo et al., 2016*). Importantly, our results have linked LOC to the representation of visual object features, and the temporal pole and parahippocampal cortex to the representation of conceptual object features. Thus, PRC has the connectivity properties that make it well suited to be a trans-modal convergence zone capable of integrating object features that are both visual and conceptual in nature. An interesting challenge for future research will be to determine how differentially attending to specific types of object features shapes functional connectivity profiles between these regions.

Although speculative, results from the current study suggest that attention may modulate information both within and between the ROIs examined. First, we see visual similarity coding in LOC only when task demands biased attention to visual object features, and conceptual similarity coding in parahippocampal cortex only when task demands biased attention to conceptual object features. Second, we saw a trend toward an interaction between behavior-based models and fMRI task context in PRC, such that visual similarity coding was more pronounced in the visual task context than was conceptual similarity coding, and vice versa. Thus, attending to specific types of features did not merely manifest as univariate gain modulation. Rather, attention appeared to modulated multi-voxel activity patterns.

Another novel aspect of our findings is that parahippocampal cortex exhibited conceptual similarity coding in the conceptual task context. Interestingly, it has been suggested that this structure broadly contributes to cognition by processing contextual associations, including the co-occurrence of objects within a context (*Bar, 2004*; *Aminoff et al., 2013*). Critically, objects that regularly co-occur in the same context (e.g. 'comb' and 'hairdryer' in a barbershop) often share many conceptual features (e.g. functional properties such as '*used to style hair*'), but do not necessarily share many visual features. Thus, object-evoked responses in parahippocampal cortex may express feature-based conceptual similarity structure because objects with many shared conceptual features bring to mind an associated context, whereas objects that are visually similar but conceptually distinct do not (e.g. hairdryer and gun). We note, however, that the current study was not designed to test-specific hypotheses about the contextual co-occurrence of objects, or how co-occurrence relates to conceptual feature statistics. Ultimately, a mechanistic account of object-based coding in PHC will require further research using a carefully selected stimulus set in which the strength of contextual associations (i.e. co-occurrence) between object concepts is not confounded with conceptual features.

In summary, this study sheds new light on our understanding of how object concepts are represented in the brain. Specifically, we revealed that PRC represented object concepts in a task-invariant, integrative similarity code that captured the visual and conceptual relatedness among stimuli. Most critically, this result was obtained despite systematically dissociating visual and conceptual features across object concepts. Moreover, the striking neuroanatomical specificity of this result suggests that PRC uniquely supports integration across these fundamentally different types of features. Ultimately, this pattern of results implicates PRC in the representation of fully-specified objects.

# Materials and methods

## Participants

### Behavior-based visual similarity rating task and conceptual feature generation task

A total of 2846 individuals completed online behavioral tasks using Amazon's Mechanical Turk (https://www.mturk.com). Data from 61 participants were discarded due to technical errors, incomplete submissions, or missed catch trials. Of the remaining 2785 participants, 1185 completed the visual similarity rating task (616 males, 569 females; age range = 18–53; mean age = 30.1), and 1600 completed the semantic feature generation task (852 males, 748 females; age range = 18–58 years; mean age = 31.7). These sample sizes are proportionally in line with those reported by *McRae et al., 2005*. Individuals who completed the visual similarity rating task were excluded from completing the feature generation task, and vice versa. All participants provided informed consent and were compensated for their time. Both online tasks were approved by the University of Toronto Ethics Review Board.

### Brain-based fMRI task

A separate group consisting of sixteen right-handed participants took part in the fMRI experiment (10 female; age range = 19–29 years; mean age = 23.1 years). This sample size is in line with extant fMRI studies that have used comparable analytical procedures to test hypotheses pertaining to object representation in the ventral visual stream and ATL (*Bruffaerts et al., 2013*; *Devereux et al., 2013*; *Martin et al., 2013*; *2016*; *2018*; *Clarke and Tyler, 2014*;*Chadwick et al., 2016*; *Erez et al., 2016*; *Borghesani et al., 2016*). Due to technical problems, we were unable to obtain data from one experimental run in two different participants. No participants were removed due to excessive motion using a criterion of 1.5 mm of translational displacement. All participants gave informed consent, reported that they were native English speakers, free of neurological and psychiatric disorders, and had normal or corrected to normal vision. Participants were compensated $50. This study was approved by the Baycrest Hospital Research Ethics Board.

## Stimuli

As a starting point, we chained together a list of 80 object concepts in such a way that adjacent items in the list alternated between being conceptually similar but visually distinct and visually similar but conceptually distinct (e.g. bullet – gun – hairdryer – comb; bullet and gun are conceptually but not visually similar, whereas gun and hairdryer are visually but not conceptually similar, and hairdryer and comb are conceptually but not visually similar, etc.). Our initial stimulus set was established using the authors' subjective impressions. The visual and conceptual similarities between all pairs of object concepts were then quantified by human observers in the context of a visual similarity rating task and a conceptual feature generation task, respectively. Results from these behavioral tasks were then used to select 40 object concepts used throughout the current study.

Participants who completed the visual similarity rating task were presented with 40 pairs of words and asked to rate visual similarity between the object concepts to which they referred (*Figure 1A*). Responses were made using a 5-point scale (very dissimilar, somewhat dissimilar, neutral, somewhat similar, very similar). Each participant was also presented with four catch trials on which an object concept was paired with itself. Across participants, 95.7% of catch trials were rated as being very similar. Data were excluded from 28 participants who did not rate all four catch trials as being at least 'somewhat similar'. Every pair of object concepts from the initial set of 80 object concepts (3160) was rated by 15 different participants.

We next quantified conceptual similarities between object concepts based on responses obtained in a conceptual feature generation task (*Figure 1B*), following task instructions previously described by *McRae et al., 2005*. Each participant was presented with one object concept and asked to produce a list of up to 15 different types of descriptive features, including functional properties (e.g. what it is used for, where it is used, and when it is used), physical properties (e.g. how it looks, sounds, smells, feels, and tastes), and other facts about it, such as the category to which it belongs or other encyclopedic facts (e.g. where it is from). One example object and its corresponding features from a normative database were presented as an example (*McRae et al., 2005*). Interpretation

and organization of written responses were guided by criteria described by *McRae et al., 2005*. Features were obtained from 20 different participants for each object concept. Data were excluded from 33 participants who failed to list any features. A total of 4851 unique features were produced across all 80 object concepts and participants. Features listed by fewer than 4 out of 20 participants were considered to be unreliable and discarded for the purpose of all subsequent analyses, leaving 723 unique features. This exclusion criterion is proportionally comparable to that used by *McRae et al., 2005*. On average, each of the 80 object concepts was associated with 10.6 features.

We used a data-driven approach to select a subset of 40 object concepts from the initial 80-item set. These 40 object concepts are reflected in the behavior-based visual and conceptual RDMs, and were used as stimuli in our fMRI experiment. Specifically, we first ensured that each object concept was visually similar, but conceptually dissimilar, to at least one other item (e.g. hairdryer – gun), and conceptually similar, but visually dissimilar, to at least one different item (e.g. hairdryer – comb). Second, in an effort to ensure that visual and conceptual features varied independently across object concepts, stimuli were selected such that the corresponding behavior-based visual and conceptual similarity models were not correlated with one another.

## Behavior-based RDMs

### Behavior-based visual RDM

A behavior-based model that captured visual dissimilarities between all pairs of object concepts included in the fMRI experiment (40 object concepts) was derived from the visual similarity judgments obtained from our online rating task. Specifically, similarity ratings for each pair of object concepts were averaged across participants, normalized, and expressed within a 40 × 40 RDM (1 − averaged normalized rating). Thus, the value in a given cell of this RDM reflects the visual similarity of the object concepts at that intersection. This behavior-based visual RDM is our visual dissimilarity model.

### Behavior-based conceptual RDM

A behavior-based model that captured conceptual dissimilarities between all pairs of object concepts included in the fMRI experiment was derived from data obtained in our online feature-generation task. In order to ensure that the semantic relationships captured by our conceptual similarity model were not influenced by verbal descriptions of visual attributes, we systematically removed features that characterized either visual form or color (e.g. 'is round' or 'is red'). Using these criteria a total of 58 features (8% of the total number of features provided) were removed. We next quantified conceptual similarity using a concept-feature matrix in which columns corresponded to object concepts (i.e. 40 columns) and rows to the conceptual features associated with those objects (i.e. 282 rows) (*Figure 1B*, center). Specifically, we computed the cosine angle between each row; cosine similarity reflects the conceptual distances between object concepts such that high cosine similarities between items denote short conceptual distance. The conceptual dissimilarities between all pairs of object concepts were expressed as a 40 × 40 RDM. The value within each cell of the conceptual model RDM was calculated as 1 − the cosine similarity value between the corresponding object concepts. This behavior-based conceptual RDM is our conceptual dissimilarity model.

### Behavior-based RSA: comparison of behavior-based RDMs

We next quantified similarity between our behavior-based visual RDM and behavior-based conceptual RDM using Kendall's tau-a as the relatedness measure. This ranked correlation coefficient is the most appropriate inferential statistic to use when comparing sparse RDMs that predict many tied ranks (i.e. both models predict complete dissimilarity between many object pairs; *Nili et al., 2014*). Statistical analysis of model similarity was performed using a stimulus-label randomization test (10,000 iterations) that simulated the null hypothesis of unrelated RDMs (i.e. zero correlation) based on the obtained variance. Significance was assessed through comparison of the obtained Kendall's tau-a coefficient to the equivalent distribution of ranked null values. As noted in the Results section, this analysis revealed that our behavior-based visual and conceptual RDMs were not significantly correlated (Kendall's tau-a = 0.01, p=0.10). Moreover, inclusion of the 58 features that described color and visual form in the behavior-based conceptual RDM did not significantly alter its relationship with the visual behavior-based visual RDM (Kendall's tau-a = 0.01, p=0.09).

## Experimental procedures: fMRI feature verification task

During scanning, participants completed a feature verification task that required a yes/no judgment indicating whether a given feature was applicable to a specific object concept on a trial-by-trial basis. We systematically varied the feature verification probes in a manner that established a visual feature verification task context and conceptual feature verification task context. Verification probes comprising the visual task context were selected to encourage processing of the visual semantic features that characterize each object concept (i.e. shape, color, and surface detail). To this end, eight specific probes were used: shape [(angular, rounded), (elongated, symmetrical)], color (light, dark), and surface (smooth, rough). Notably, all features are associated with two opposing probes (e.g. angular and rounded; natural and manufactured) to ensure that participants made an equal number of 'yes' and 'no' responses. Verification probes comprising the conceptual feature verification task context were selected to encourage processing of the abstract conceptual features that characterize each object concept (i.e. animacy, origin, function, and affective associations). To this end, eight specific verification probes were used: (living, non-living), (manufactured, natural), (tool, non-tool), (pleasant, unpleasant).

## Procedures

The primary experimental task was evenly divided over eight runs of functional data acquisition. Each run lasted 7 m 56 s and was evenly divided into two blocks, each of which corresponded to either a visual verification task context or a conceptual feature verification task context. The order of task blocks was counter-balanced across participants. Each block was associated with a different feature verification probe, with the first and second block in each run separated by 12 s of rest. Blocks began with an 8 s presentation of a feature verification probe that was to be referenced for all intra-block trials. With this design, each object concept was repeated 16 times: eight repetitions across the visual feature verification task context and eight repetitions across the conceptual feature verification task context. Behavioral responses were recorded using an MR-compatible keypad.

Stimuli were centrally presented for 2 s and each trial was separated by a jittered period of baseline fixation that ranged 2–6 s. Trial order and jitter interval were optimized for each run using the OptSeq2 algorithm (http://surfer.nmr.mgh.harvard.edu/optseq/), with unique sequences and timing across counterbalanced versions of the experiment. Stimulus presentation and timing was controlled by E-Prime 2.0 (Psychology Software Tools, Pittsburgh, PA).

## Experimental procedure: fMRI functional localizer task

Following completion of the main experimental task, each participant completed an independent functional localizer scan that was subsequently used to identify LOC. Participants viewed objects, scrambled objects, words, scrambled words, faces, and scenes in separate 24 s blocks (12 functional volumes). Within each block, 32 images were presented for 400 ms each with a 350 ms ISI. There were four groups of six blocks, with each group separated by a 12 s fixation period, and each block corresponding to a different stimulus category. Block order (i.e. stimulus category) was counterbalanced across groups. All stimuli were presented in the context of a 1-back task to ensure that participants remained engaged throughout the entire scan. Presentation of images within blocks was pseudo-random with 1-back repetition occurring 1–2 times per block.

## ROI definitions

We performed RSA in four a priori defined ROIs. The temporal pole, PRC, and parahippocampal cortex were manually defined in both the left and right hemisphere on each participant's high-resolution anatomical image according to established MR-based protocols (*Pruessner et al., 2002*), with adjustment of posterior border of parahippocampal cortex using anatomical landmarks described by *Frankó et al. (2014)*. Lateral occipital cortex was defined as the set of contiguous voxels located along the lateral extent of the occipital lobe that responded more strongly to intact than scrambled objects (p<0.001, uncorrected; *Malach et al., 1995*).

## fMRI data acquisition

Scanning was performed using a 3.0 T Siemens MAGNETOM Trio MRI scanner at the Rotman Research Institute at Baycrest Hospital using a 32-channel receiver head coil. Each scanning session

began with the acquisition of a whole-brain high-resolution magnetization-prepared rapid gradient-echo T1-weighted structural image (repetition time = 2 s, echo time = 2.63 ms, flip angle = 9°, field of view = 25.6 cm², 160 oblique axial slices, 192 × 256 matrix, slice thickness = 1 mm). During each of eight functional scanning runs comprising the main experimental task, a total of 238 T2*-weighted echo-planar images were acquired using a two-shot gradient echo sequence (200 × 200 mm field of view with a 64 × 64 matrix size), resulting in an in-plane resolution of 3.1 × 3.1 mm for each of 40 2 mm axial slices that were acquired in an interleaved manner along the axis of the hippocampus. The inter-slice gap was 0.5 mm; repetition time = 2 s; echo time = 30 ms; flip angle = 78°). These parameters yielded coverage of the majority of cortex, excluding only the most superior aspects of the frontal and parietal lobes. During a single functional localizer scan, a total of 360 T2*-weighted echo-planar images were acquired using the same parameters reported for the main experimental task. Lastly, a B0 field map was collected following completion of the functional localizer scan

## fMRI data analysis software

Preprocessing and GLM analyses were performed in FSL5 (*Smith et al., 2004*). Representational similarity analyses were performed using CoSMoMVPA (http://www.cosmomvpa.org/; *Oosterhof et al., 2016*).

## Preprocessing and estimation of object-specific multi-voxel activity patterns

Images were initially skull-stripped using a brain extraction tool (BET, *Smith, 2002*) to remove non-brain tissue from the image. Data were then corrected for slice-acquisition time, high-pass temporally filtered (using a 50-s period cut-off for event-related runs, and a 128 s period cut-off for the blocked localizer run), and motion corrected (MCFLIRT, *Jenkinson et al., 2002*). Functional runs were registered to each participant's high-resolution MPRAGE image using FLIRT boundary-based registration with B0-fieldmap correction. The resulting unsmoothed data were analyzed using first-level FEAT (v6.00; fsl.fmrib.ox.ac.uk/fsl/fslwiki) in each participant's native anatomical space. Parameter estimates of BOLD response amplitude were computed using FILM, with a general linear model that included temporal autocorrelation correction and six motion parameters as nuisance covariates. Each trial (i.e. object concept) was modeled with a delta function corresponding to the stimulus presentation onset and then convolved with a double-gamma hemodynamic response function. Separate response-amplitude (β) images were created for each object concept (n = 40), in each run (n = 8), in each property verification task context (n = 2). Obtained β images were converted into *t*-statistic maps; previous research has demonstrated a modest advantage for *t*-maps over β images in the context of multi-voxel pattern analysis (*Misaki et al., 2010*). In a final step, we created mean object-specific *t*-maps by averaging across runs. These data were used for all subsequent similarity analyses.

## Representational similarity analysis (RSA)

### ROI-based first-level RSA

We used linear correlations to quantify the participant-specific dissimilarities (1 – Pearson's r) between all object-evoked multi-voxel activity patterns (n = 40) within each ROI (n = 4). Participant-specific dissimilarity measures were expressed in 40 × 40 RDMs for each verification task context (n = 2), separately. Thus, for each ROI, each participant had one RDM that reflected the dissimilarity structure from the visual feature verification task context (i.e. brain-based visual task RDM), and one RDM that reflected the dissimilarity structure from the conceptual verification task context (i.e., brain-based conceptual task RDM).

### ROI-based second-level RSA

We performed second-level RSAs, that is, we compared RDMs derived from first-level RSAs, to quantify similarities among behavior-based RDMs and brain-based RDMs. Similarity was quantified in each participant using the ranked correlation coefficient (Kendall's tau-a) between RDMs. Inferential statistical analyses were performed using a one-sided Wilcoxon signed-rank test across subject-specific RDM correlations to test for significance. This non-parametric test provides valid inference and treats the variation across subjects as a random effect, thus supporting generalization of results

beyond the sample. A Bonferroni correction was applied in each analysis to compensate for the number of second-level comparisons.

## Searchlight-based RSA

Whole-volume RSA was implemented using 100-voxel surface-based searchlights (*Kriegeskorte et al., 2006*; *Oosterhof et al., 2011*). Each surface-based searchlight referenced the 100 nearest voxels to the searchlight center based on geodesic distance on the cortical surface. Neural estimates of dissimilarity (i.e. RDMs) were calculated in each searchlight using the same approach implemented in our ROI-based RSA. Correlations between behavior-based RDMs were also quantified using the same approach. The correlation coefficients obtained between behavior-based RDMs and brain-based RDMs were then Fisher-*z* transformed and mapped to the voxel at the centre of each searchlight to create a whole-brain similarity map. Participant-specific similarity maps were then normalized to a standard MNI template using FNIRT (*Greve and Fischl, 2009*). To assess the statistical significance of searchlight maps across participants, all maps were corrected for multiple comparisons without choosing an arbitrary uncorrected threshold using threshold-free cluster enhancement (TFCE) with a corrected statistical threshold of p<0.05 on the cluster level (*Smith and Nichols, 2009*). A Monte Carlo simulation permuting condition labels was used to estimate a null TFCE distribution. First, 100 null searchlight maps were generated for each participant by randomly permuting condition labels within each obtained searchlight RDM. Next, 10,000 null TFCE maps were constructed by randomly sampling from these null data sets in order to estimate a null TFCE distribution (*Stelzer et al., 2013*). The resulting surface-based statistically thresholded *z*-score were projected onto the PALS-B12 surface atlas in CARET version 5.6. (http://www.nitrc.org/projects/caret/; *Van Essen et al., 2001*; *Van Essen, 2005*).

## Acknowledgements

This work was supported by the Canadian Natural Sciences Engineering Research Council (Discovery and Accelerator Grants to MDB.; Postdoctoral Fellowship award to CBM), the James S. McDonnell Foundation (Scholar Award to MDB), and the Canada Research Chairs Program (MDB).

## Additional information

### Funding

| Funder | Grant reference number | Author |
| --- | --- | --- |
| Natural Sciences and Engineering Research Council of Canada | | Morgan Barense |
| James S. McDonnell Foundation | | Morgan Barense |
| Canada Research Chairs | | Morgan Barense |
| Ontario Ministry of Economic Development and Innovation | | Morgan Barense |
| Natural Sciences and Engineering Research Council of Canada | PDF - 502437 - 2017 | Chris B Martin |

The funders had no role in study design, data collection and interpretation, or the decision to submit the work for publication.

### Author contributions

Chris B Martin, Conceptualization, Data curation, Formal analysis, Investigation, Visualization, Methodology, Writing—original draft, Project administration, Writing—review and editing; Danielle Douglas, Rachel N Newsome, Conceptualization, Resources, Writing—review and editing; Louisa LY Man, Data curation, Formal analysis, Investigation; Morgan D Barense, Conceptualization, Supervision, Funding acquisition, Methodology, Project administration, Writing—review and editing

**Author ORCIDs**

Chris B Martin ⓘ http://orcid.org/0000-0002-7014-4371

**Ethics**

Human subjects: The study was approved by the Institutional Review Board at the University of Toronto (REB # 23778) and the Research Ethics Board at Baycrest Hospital (REB # 15-06). Informed consent was obtained from each participant before the experiment, including consent to publish anonymized results.

**Decision letter and Author response**

Decision letter https://doi.org/10.7554/eLife.31873.024
Author response https://doi.org/10.7554/eLife.31873.025

# Additional files

**Supplementary files**

• Transparent reporting form
DOI: https://doi.org/10.7554/eLife.31873.021

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
