## [Decision Letter]

Thank you for submitting your article "Integrative and distinctive coding of perceptual and conceptual object features in the ventral visual stream" for consideration by *eLife*. Your article has been reviewed by three peer reviewers, and the evaluation has been overseen by a Reviewing Editor and Timothy Behrens as the Senior Editor. The following individuals involved in review of your submission have agreed to reveal their identity: Anna C Schapiro (Reviewer #2).

The reviewers have discussed the reviews with one another and the Reviewing Editor has drafted this decision to help you prepare a revised submission.

This paper aims to uncover where in the brain perceptual and conceptual information is integrated. The authors compare behavioral and neural similarity amongst objects processed in visual and conceptual task contexts. They find that visual structure is represented in LOC, conceptual structure is represented in the temporal pole and parahippocampal cortex, and that perirhinal cortex is uniquely sensitive to both perceptual and conceptual information, across task contexts, suggesting that this is where conceptual and perceptual information is integrated.

The reviewers commented on the elegant nature of the study design, the stringent analysis, and how clean results were in the context of brain areas that are often difficult to get signals from (with fMRI). The paper was judged to be of broad interest and impactful.

Essential revisions:

The following issues were highlighted as items that must be addressed before publication.

1) It looks like the statistics were computed using objects – not participants – as the random effects factor, which makes it unclear if we can generalize the findings to the population. The correlation between behavioral and neural RDMs could be calculated for each individual, and then statistical testing could be done across these 16 (fisher-transformed) values. Or it may make sense to run the permutation test within each individual and then compute group statistics across the 16 zscores of the individuals relative to their own null distributions. The object-based statistics are useful, but additionally reporting these participant-based statistics will allow the reader to understand whether these findings are likely to generalize to new participants.

2) These analyses tell us about how relationships between objects are represented similarly or differently in different task contexts, but it seems like it would be useful to also report how similarly the objects themselves are represented in different task contexts. In other words, in an RDM with visual brain response as rows and conceptual as columns, does perirhinal (and perhaps temporal pole) have the strongest diagonal in that matrix?

3) There looks to be a strong interaction effect in perirhinal cortex, with its patterns of activity showing more similarity to the behavior-based visual RDM vs. the behavior-based conceptual RDM when the task is visual, and vice versa when the task is conceptual. Could the authors assess this interaction? If the authors are concerned about the assumptions of an ANOVA being violated, then perhaps a non-parametric test of an interaction can be used. The presence of an interaction does not in any way contradict what the authors are stating, but would instead add to it by suggesting that, on top of PRC representing both conceptual and perceptual information regardless of task, there is a small modulation by the task state or attentional state.

4) There is something a little odd about the depiction of the visual and conceptual RDMs in Figure 1. Why are there vertical "streaks" though the columns? Shouldn't these RDMs be completely symmetrical? Perhaps this is an artifact of the normalization procedure, which is mentioned in the methods but not described in detail. As a general point, although it is fine to scale the RDMs to use the full light/dark range, I think that it would give a better idea of the similarity space if the actual values of the similarity measures were used rather than percentile scores. Also, in the uploaded RDMs, it is puzzling to me that the values for dissimilar pairs are all 1 for the visual RDM, as this was based on a 5-point Likert scale. Surely the same value wasn't obtained for every object pairing.

5) Although I think that the procedure used to obtain behavioral measures of conceptual and perceptual similarity is a strength of the paper, it would be useful to know how these measures compare to other measures of conceptual similarity, like WordNet distances, or text corpus co-occurrence, and whether similar results are obtained when these other measures are used.

6) Although the paper focuses primarily on the ROIs, the results of the searchlight analysis will be of considerable interest to many readers and deserve more emphasis. I suggest that the brains in Figure 7A and 7B be made larger (perhaps by removing the RDMs, which take up considerable space but aren't really essential). In addition, it would be useful to plot the borders of the ROIs as outlines on the brains so that the consistency between the ROI boundaries and the searchlight analyses could be visually assessed. (As an aside, the fact that PRC is the only area showing overlap between the visual and conceptual effects in the searchlight analysis is very impressive and really underscores the strength of the effect.)

7) The results in PHC are very interesting, but the use of the PHC ROI is not strongly justified in the paper, and the implications of this result are not discussed at all. Although the paper states that PHC has been implicated by previous work on conceptual knowledge, the papers that are referenced (by Bar and Aminoff, and Ranganath) discussed a very specific kind of conceptual knowledge: knowledge about co-occurrence of objects within the same context. In my opinion, an important aspect of the current results is that they offer an important new data point in support of this idea. One possible interpretation of the PHC results is that participants bring to mind a contextual setting for the object when they perform the conceptual tasks but not when they perform the perceptual tasks. For example, "comb" and "hairdryer" might bring to mind a bathroom or a barbershop, but only when thinking about the conceptual meaning of these objects, not when thinking about their shape or color.

8) It would be useful to have more precise information about the locus of the PHC effect relative to the "parahippocampal place area" (PPA), which tends to extend posterior of PHC proper. Several papers suggest an anterior/posterior division within the PPA whereby the more anterior portions represent more abstract/conceptual information and the more posterior portions represent more visual/spatial information (e.g. Baldassano et al., 2013; Marchette et al., 2015; Aminoff et al., 2006). The functional localizer includes both scenes and objects so it should be possible for the authors to identify the PPA, and thus report whether their effects are in PPA or not, and if so, if they are in the more anterior portion.

9) I wonder if the authors could comment on whether they think the results would change if pictures, rather than words, were used for the objects in the fMRI experiment. For example, might LOC represent visual similarity structure in a task-invariant way if pictures are presented, because their visual features would be processed more automatically than if words are presented, and visual features of the objects have to be brought to mind? I was also curious if the authors could explain their motivation for using words rather than images – was it to force participants to bring to mind both visual and conceptual information?

10) It is notable that conceptual effects were limited to PRC, PHC, and TP, and were not found in other regions of the brain. What implications does this have in light of previous work that has found a wider distribution of conceptual regions? For example, Fairhall and Caramazza (2013) identified "amodal" conceptual processing in inferior temporal gyrus, posterior cingulate, angular gyrus, prefrontal regions? What about the ventral stream regions outside of LO, PRC, and PHC, like the fusiform gyrus? (I'm not actually suggesting that the authors use these regions as ROIs, but more focus on the whole-brain results and comparison to these earlier findings would be recommended.)

[Editors' note: further revisions were requested prior to acceptance, as described below.]

Thank you for resubmitting your work entitled "Integrative and distinctive coding of visual and conceptual object features in the ventral visual stream" for further consideration at *eLife*. Your revised article has been favorably evaluated by Timothy Behrens (Senior Editor), a Reviewing Editor, and three reviewers.

The manuscript has been improved but there are some remaining issues that need to be addressed before acceptance, as outlined below:

1) It was a great idea to put the ROI boundaries on the brains depicting the searchlight results. I'd recommend making the ROI borders a bit thicker, though, and perhaps changing their colors, to make them easier to see. The perirhinal and parahippocampal ROIs are fairly clear, but the LOC and temporal pole less so.

2) Now that the ROI outlines have been added to the searchlight results, it is apparent (Figure 10B) that the locus of conceptual decoding in parahippocampal cortex straddles the perirhinal/parahippocampal border. This fact is perhaps worth mentioning in the text. At present, the overlap with anterior PPA is emphasized, but one might equally emphasize that the overlap with PPA – and even PHC – is only partial.

3) Also, there seems to be an error in the caption for Figure 10B, which is described as depicting correlation between brain-based VISUAL and behavior-based conceptual RDMs – both should be conceptual. The label on the figure itself is different (and presumably correct).

---

## [Author Response]

Essential revisions:The following issues were highlighted as items that must be addressed before publication.1) It looks like the statistics were computed using objects – not participants – as the random effects factor, which makes it unclear if we can generalize the findings to the population. The correlation between behavioral and neural RDMs could be calculated for each individual, and then statistical testing could be done across these 16 (fisher-transformed) values. Or it may make sense to run the permutation test within each individual and then compute group statistics across the 16 zscores of the individuals relative to their own null distributions. The object-based statistics are useful, but additionally reporting these participant-based statistics will allow the reader to understand whether these findings are likely to generalize to new participants.

We thank the reviewers for this important suggestion, and agree that reporting results from participant-based random-effects analyses will provide a more complete understanding of our data. As such, we now report results from inferential statistical analyses, using participants as a random factor, for all ROI-based aspects of our study. Source data files containing participant-specific correlations are now linked to each ROI-based results figure. We tested for significant positive correlations in each analysis by calculating participant-specific similarity indices (Kendall’s tau-a) between RDMs, which were then tested across using a one-sided Wilcoxon signed-rank test. This non-parametric test provides valid inference and treats the variation across participants as a random effect, thus supporting generalization of results beyond the sample (Nili et al., 2014).

Our new results are described throughout the Results section of our revised manuscript. In short, the overall pattern of results revealed by our random-effects analyses was not meaningfully different from that reported in our initial submission. Specifically, we still obtain evidence for visual similarity coding in LOC, conceptual similarity coding in the temporal pole and parahippocampal cortex, and both visual and conceptual similarity coding in PRC. Given the similarities in results across statistical procedures, together with an interest in avoiding confusion, we have replaced (rather than augmented) all fixed-effects results with these statistically more stringent random-effects results.

2) These analyses tell us about how relationships between objects are represented similarly or differently in different task contexts, but it seems like it would be useful to also report how similarly the objects themselves are represented in different task contexts. In other words, in an RDM with visual brain response as rows and conceptual as columns, does perirhinal (and perhaps temporal pole) have the strongest diagonal in that matrix?

We have addressed this theoretically important question using the analytical approach recommended by the reviewers (i.e., a first-level RSA). All pertinent results are reported in a new paragraph titled “ROI-Based RSA: Comparisons of Within-Object Multi-Voxel Activity Patterns Across Different Task Contexts”. Specifically, we first calculated one dissimilarity value (1 – Pearson’s *r*) between the mean multi-voxel activity patterns evoked by a given object concept across different task contexts. These 40 within-object dissimilarity values were expressed along the diagonal of an RDM for each ROI in each participant (Figure 8A). We next calculated mean within-object dissimilarity (i.e., the mean of the diagonal in the RDM) for the purpose of performing statistical inference.

Results are presented in Figure 8B. Within-object similarity did not differ from zero in either LOC (Pearson’s r =.007, p =.20) or parahippocampal cortex (Pearson’s r = -.008, p =.87), suggesting that a given object concept was represented differently across the visual and conceptual task contexts in these ROIs. These findings are consistent with the task-dependent nature of the similarity codes we observed in these regions (Figures 6A, 6B). Conversely, within-object similarity was significantly greater than zero in PRC (Pearson’s r = 0.41, p <.01, Bonferroni corrected for four comparisons), and the temporal pole (Pearson’s r = 0.34, p <.05, Bonferroni corrected for four comparisons). These results indicate that a given object concept was represented similarly across task contexts in these structures. In the temporal pole, this reflected the fact that conceptual object information (e.g., “used to style hair” and “found in salons”) was emphasized across task contexts (Figure 6D). By contrast, we view high within-object similarity in PRC as further evidence of integrative coding, as this structure appeared to carry conceptual (e.g., “used to style hair” and “found in salons”) and visual (e.g., visually similar to a gun) object information in both task contexts (Figure 6C).

3) There looks to be a strong interaction effect in perirhinal cortex, with its patterns of activity showing more similarity to the behavior-based visual RDM vs. the behavior-based conceptual RDM when the task is visual, and vice versa when the task is conceptual. Could the authors assess this interaction? If the authors are concerned about the assumptions of an ANOVA being violated, then perhaps a non-parametric test of an interaction can be used. The presence of an interaction does not in any way contradict what the authors are stating, but would instead add to it by suggesting that, on top of PRC representing both conceptual and perceptual information regardless of task, there is a small modulation by the task state or attentional state.

The notion of task-based modulation is an interesting proposition that has been widely discussed in the context of representational accounts of PRC functioning (see Graham et al., 2010 for review). Here, we addressed this essential issue by first converting Kendall’s tau-a values obtained in PRC to Pearson’s *r* coefficients (*r* = sin (½ π tau-a)), which were then Fisher-*z* transformed. These transformations were guided by procedures developed by Walker (2003). In line with the reviewers’ suspicion, a 2 (task context) x 2 (behaviour-based model) repeated-measures ANOVA revealed a behavior-based model x fMRI task context task interaction that neared, but did not reach, significance (F(1,15) = 3.48, p =.082). This result is now reported in our revised our manuscript (subsection “Perirhinal Cortex Represents Object Concepts in a Task-Invariant Similarity Code that Reflects Integration of Visual and Conceptual Features”, second paragraph).

4) There is something a little odd about the depiction of the visual and conceptual RDMs in Figure 1. Why are there vertical "streaks" though the columns? Shouldn't these RDMs be completely symmetrical? Perhaps this is an artifact of the normalization procedure, which is mentioned in the methods but not described in detail. As a general point, although it is fine to scale the RDMs to use the full light/dark range, I think that it would give a better idea of the similarity space if the actual values of the similarity measures were used rather than percentile scores. Also, in the uploaded RDMs, it is puzzling to me that the values for dissimilar pairs are all 1 for the visual RDM, as this was based on a 5-point Likert scale. Surely the same value wasn't obtained for every object pairing.

We are grateful to the reviewers for noting the asymmetries in our depictions of the behaviour-based RDMs. This was indeed an artifact of the normalization procedure. The issue has been rectified in our revised manuscript, where we now depict raw dissimilarity values (1 – Kendall’s tau-a), rather than percentile scores, in Figure 1.

With respect to values in the behaviour-based visual RDM, it is the case that the majority of object pairs were unanimously (N = 15) endorsed with the lowest similarity rating. For example, all participants responded “not at all visually similar (response option 1)” when asked to rate the visual similarity between antlers – hairdryer, gun – pillow, and pen – compass. On average, this was the case for the relationship between a given object concept and 35.95 out of 40 other object concepts. This pattern reflects the result of a deliberate data-driven approach to select a stimulus set in which visual and conceptual similarities were not confounded across objects. Beginning with an initial stimulus set of 80 items, we systematically excluded object concepts that were visually similar (even to a small degree) to many other objects. Similar criteria were used with respect to conceptual similarity, resulting in an average of 35.2 pairs of object concepts for which there were zero common conceptual features. Selecting stimuli that yielded sparse behaviour-based RDMs was necessary to ensure that conceptual similarities did not correlate with visual similarities. Ultimately, we view the sparsity of our behavior-based RDMs as a strength of our experimental design.

5) Although I think that the procedure used to obtain behavioral measures of conceptual and perceptual similarity is a strength of the paper, it would be useful to know how these measures compare to other measures of conceptual similarity, like WordNet distances, or text corpus co-occurrence, and whether similar results are obtained when these other measures are used.

The reviewer has raised an important question regarding the degree to which our results generalize across conceptual similarity models obtained using different procedures. To address this issue, we have implemented a word2vec language model, which mapped 3 million words to 300 feature vectors in a high-dimensional space. The model was trained using ~100 billion words from a Google News dataset. From this model we calculated the cosine similarity (i.e., semantic similarity) between feature vectors for all pairs of words in our stimulus set. These data were expressed in a 40x40 word2vec RDM. The word2vec model is now included in Figure 1—source data 1.

Importantly, the word2vec RDM was significantly correlated with our behavior-based conceptual RDM (Kendall’s tau-a =.11, SE =.0141, *p* <.00001), suggesting that both models capture the conceptual similarity structure among the object concepts. However, the word2vec RDM was also significantly correlated with our behavior-based visual RDM (Kendall’s tau-a =.04, SE =.0130, *p* <.001). This result suggests that, whereas our behavior-based conceptual RDM captured semantic similarity as it is narrowly defined as conceptual object features, the word2vec RDM may have captured a broader definition of semantic similarity, i.e., one that includes both visual semantics and abstract conceptual features. Consistent with this view, gun and hairdryer were conceptually unrelated in our behavior-based conceptual RDM (cosine = 0), whereas the word2vec RDM suggested modest conceptual similarity (cosine =.16). Although this difference is likely determined by multiple factors, it is important to note that gun and hairdryer had a relatively high visual similarity index in our behaviour-based visual RDM (normalized mean rating =.58). Ultimately, these data highlight a theoretically important distinction between behaviorally-derived conceptual feature-based statistics and corpus-based estimates of semantic similarity. We report comparisons between our behavior-based RDMs and the word2vec RDM in a new section titled “Comparison of Behavior-Based RDMs with a Corpus-Based Semantic RDM*”*.

For the purpose of comparison, we next compared the word2vec RDM with brain-based RDMs using the same procedures described in the previous section. Results are presented in Figure 6—figure supplement 1. These analyses revealed significant positive correlations between the word2vec RDM and the brain-based conceptual task RDMs in parahippocampal cortex (Kendall’s tau-a =.05, *p* <.01), PRC (Kendall’s tau-a =.035, *p* <.01), and the temporal pole (Kendall’s tau-a =.029, *p* <.01). The word2vec RDM was also significantly correlated with the brain-based visual task RDMs in PRC (Kendall’s tau-a =.025, *p* <.05) and the temporal pole (Kendall’s tau-a =.027, *p* <.05). Thus, the pattern of results obtained using the word2vec RDM was identical to that obtained using the behavior-based conceptual RDMs in parahippocampal cortex, PRC, and the temporal pole. Interestingly, however, the word2vec RDM was also significantly correlated with the brain-based visual task RDMs in LOC (Kendall’s tau-a =.028, *p* <.05). This result is consistent with the observation that the word2vec RDM was significantly correlated with our behavior-based visual RDM, and further suggests that corpus-based models of semantic memory likely capture similarities between object concepts at the level of abstract conceptual properties and visual semantics. These results are reported in a new section titled “ROI-Based RSA: Comparison of Corpus-Based (word2vec) Semantic RDM with Brain-Based RDMs”.

6) Although the paper focuses primarily on the ROIs, the results of the searchlight analysis will be of considerable interest to many readers and deserve more emphasis. I suggest that the brains in Figure 7A and 7B be made larger (perhaps by removing the RDMs, which take up considerable space but aren't really essential). In addition, it would be useful to plot the borders of the ROIs as outlines on the brains so that the consistency between the ROI boundaries and the searchlight analyses could be visually assessed. (As an aside, the fact that PRC is the only area showing overlap between the visual and conceptual effects in the searchlight analysis is very impressive and really underscores the strength of the effect.)

We thank the reviewers for this suggestion. We have increased the size of the searchlight similarity maps and deconstructed what was previously a composite figure. Results are now presented across three separate figures: similarity maps from the visual task context in Figure 9, from the conceptual task context in Figure 10, and similarity map overlap in Figure 11. Moreover, we now project the borders of each ROI onto the brains. Thanks to this advice, the neuroanatomical specificity of our results is now particularly apparent.

7) The results in PHC are very interesting, but the use of the PHC ROI is not strongly justified in the paper, and the implications of this result are not discussed at all. Although the paper states that PHC has been implicated by previous work on conceptual knowledge, the papers that are referenced (by Bar and Aminoff, and Ranganath) discussed a very specific kind of conceptual knowledge: knowledge about co-occurrence of objects within the same context. In my opinion, an important aspect of the current results is that they offer an important new data point in support of this idea. One possible interpretation of the PHC results is that participants bring to mind a contextual setting for the object when they perform the conceptual tasks but not when they perform the perceptual tasks. For example, "comb" and "hairdryer" might bring to mind a bathroom or a barbershop, but only when thinking about the conceptual meaning of these objects, not when thinking about their shape or color.

We agree that we should have more explicitly motivated the inclusion of PHC as an ROI. Accordingly, we have revised our Introduction to provide rationale for targeting PHC (and all other ROIs). Specifically, we describe the proposal that the PHC is necessary for processing contextual associations, including the co-occurrence of objects in a common context (e.g., “comb” and “hairdryer” in a barbershop; Aminoff et al., 2013). We reason that objects that are regularly encountered in the same context often share many conceptual features (e.g., “used to style hair”). To the extent that shared conceptual features directly shape contextual meaning, object-evoked responses in parahippocampal cortex may express conceptual similarity structure.

In our revised Discussion (seventh paragraph), we now consider the implications of our findings in PHC for pertinent theoretical models. We agree that the correlation we revealed between our behaviour-based conceptual RDM and brain-based conceptual task RDMs could be interpreted as support for the notion that PHC represents contextual associations. We note, however, that the current study was not designed to test specific hypotheses about the contextual co-occurrence of objects, or how co-occurrence relates to conceptual feature statistics. Ultimately, a mechanistic account of object-based coding in PHC will require further research using a carefully selected stimulus set in which the strength of contextual associations (i.e., co-occurrence) between object concepts is not confounded with conceptual features.

8) It would be useful to have more precise information about the locus of the PHC effect relative to the "parahippocampal place area" (PPA), which tends to extend posterior of PHC proper. Several papers suggest an anterior/posterior division within the PPA whereby the more anterior portions represent more abstract/conceptual information and the more posterior portions represent more visual/spatial information (e.g. Baldassano et al., 2013; Marchette et al., 2015; Aminoff et al., 2006). The functional localizer includes both scenes and objects so it should be possible for the authors to identify the PPA, and thus report whether their effects are in PPA or not, and if so, if they are in the more anterior portion.

We have conducted a new analysis to better understand how conceptual coding in PHC relates to PPA. Specifically, we first identified PPA in individual participants using whole-brain univariate GLMs (scenes > objects, liberal statistical threshold of *p* <.01 uncorrected). Using this approach, we localized a contiguous cluster of voxels that extended from posterior PHC into lingual gyrus in 14/16 participants. We next identified group-level PPA ROIs by performing a whole-brain random-effects univariate analysis with these 14 participants (FDR *p* <.05; right PPA peak MNI coordinates 22, -41, -13, cluster extant of 544 voxels with 27% overlapping posterior PHC; left PPA peak MNI coordinates -23, -44, -12, cluster extant of 512 voxels with 29% overlap with posterior PHC). Interestingly, the anterior extant of the right PPA ROI did overlap (26 voxels) with the cluster of voxels in right PHC that showed evidence of conceptual coding in the conceptual task (64 voxels). Thus, 41% of the conceptual coding cluster in PHC was situated within PPA. However, this set of voxels represented only 5% of PPA. This result is now reported in our revised Results section (subsection “Perirhinal Cortex is the Only Cortical Region that Supports Integrative Coding of Conceptual and Visual Object Features”, second paragraph).

9) I wonder if the authors could comment on whether they think the results would change if pictures, rather than words, were used for the objects in the fMRI experiment. For example, might LOC represent visual similarity structure in a task-invariant way if pictures are presented, because their visual features would be processed more automatically than if words are presented, and visual features of the objects have to be brought to mind? I was also curious if the authors could explain their motivation for using words rather than images – was it to force participants to bring to mind both visual and conceptual information?

The question of whether our results would change if pictures were used is very interesting, in particular as it relates to integrative coding in PRC. First, our rationale for using words, rather than images, in the current study reflects our specific interest in understanding pre-existing representations of object concepts, rather than bottom-up perceptual processing. Within this context, using words ensured that conceptual and visual features were extracted from pre-existing representations of object concepts. That is to say, both conceptual and visual features were arbitrarily related to the physical input (i.e., the orthography of the word). We are now explicit about this point in our revised manuscript (subsection “fMRI Task and Behavioral Results”, first paragraph).

We are actively conducting follow-up research that investigates where in the brain the content of bottom-up perceptual object processing interfaces with corresponding abstract conceptual knowledge. One challenge associated with using pictures is identifying prototypical pictorial exemplars that do not have idiosyncratic visual properties that bias attention toward individual features and away from the multiple features that together lend themselves to conceptual meaning. Moreover, using pictures may introduce systematic differences between decision making strategies in the visual and conceptual verification task contexts. For example, when pictures are used as stimuli, visual features are accessible from the pictorial cue, whereas conceptual features require abstraction from the cue.

These issues notwithstanding, our prediction is that using pictures would reveal task-invariant visual similarity coding in LOC, in line with the reviewers’ suggestion. In PHC, we would still predict conceptual similarity coding, but we do not have strong predictions about whether using pictures would influence task specificity. We would not anticipate any significant differences between words and pictures in the temporal pole, as results from previous research, including that conducted with semantic dementia, have linked this structure to conceptual coding irrespective of input modality (Patterson et al., 2007). Most importantly, we predict that PRC would show evidence of integrative coding regardless of whether words or pictures are used as stimuli.

10) It is notable that conceptual effects were limited to PRC, PHC, and TP, and were not found in other regions of the brain. What implications does this have in light of previous work that has found a wider distribution of conceptual regions? For example, Fairhall and Caramazza (2013) identified "amodal" conceptual processing in inferior temporal gyrus, posterior cingulate, angular gyrus, prefrontal regions? What about the ventral stream regions outside of LO, PRC, and PHC, like the fusiform gyrus? (I'm not actually suggesting that the authors use these regions as ROIs, but more focus on the whole-brain results and comparison to these earlier findings would be recommended.)

We agree that it is important to situate our results into the context of related studies. To this end, we first note that previous fMRI studies of semantic memory have reliably identified a network of cortical regions that contribute to semantic memory decisions, including inferior frontal gyrus, superior frontal gyrus, middle temporal gyrus, ventromedial prefrontal cortex, fusiform gyrus, the posterior cingulate, supramarginal gyrus, and the angular gyrus (Binder et al., 2009). Interestingly, when we investigated brain regions more strongly activated by our conceptual task context relative to our visual task context using a whole-brain univariate GLM analysis (FDR p <. 05), we found a set of regions that converges with this network. Namely, we see differential conceptual task involvement in inferior frontal gyrus, fusiform gyrus, middle temporal gyrus, and inferior temporal gyrus (Author Response Image 1).

Although these consistencies between the current study and previous research pertaining to semantic memory are encouraging, they do not speak to the important question of whether a given region actually represents semantic content. Rather, this is typically inferred when representational distinctions captured by conceptual similarity models map onto corresponding distinctions among object-specific neural representations. Results from RSA-based studies that have used this approach tend to reveal semantic similarity coding in a relatively limited number of brain regions. For example, Bruffaerts et al., (2013), and Clarke and Tyler (2014) found semantic similarity effects in PRC only. Related studies point to the temporal pole as a region that supports conceptual similarity coding (Peelen and Caramazza, 2012; Chadwick et al., 2016). Lastly, an examination of semantic similarities between stimulus categories found evidence for object-based similarity coding in only middle temporal gyrus and the posterior cingulate (Fairhall and Caramazza (2013).

Against this background, we are not surprised by the fact that we revealed conceptual similarity coding in only three brain regions. Further, we emphasize that our effects in PRC and the temporal pole are consistent with those reported by Bruffaerts et al. (2013), Clarke and Tyler (2014), Peelen and Caramazza (2012), and Chadwick et al. (2016). We now acknowledge these consistencies in our manuscript where we report searchlight-based RSA results (subsection “Perirhinal Cortex is the Only Cortical Region that Supports Integrative Coding of Conceptual and Visual Object Features”).

[Editors' note: further revisions were requested prior to acceptance, as described below.]

The manuscript has been improved but there are some remaining issues that need to be addressed before acceptance, as outlined below:1) It was a great idea to put the ROI boundaries on the brains depicting the searchlight results. I'd recommend making the ROI borders a bit thicker, though, and perhaps changing their colors, to make them easier to see. The perirhinal and parahippocampal ROIs are fairly clear, but the LOC and temporal pole less so.We agree that the ROI boundaries could be more salient. Accordingly, we have increased the thickness of all borders in Figures 9, 10, and 11. We have also slightly modified some colors to improve visibility.2) Now that the ROI outlines have been added to the searchlight results, it is apparent (Figure 10B) that the locus of conceptual decoding in parahippocampal cortex straddles the perirhinal/parahippocampal border. This fact is perhaps worth mentioning in the text. At present, the overlap with anterior PPA is emphasized, but one might equally emphasize that the overlap with PPA – and even PHC – is only partial.

The point regarding the neuroanatomical specificity of the searchlight result in question is well taken. As such, we now describe the cluster in which we find evidence of conceptual similarity coding in the conceptual task context as being situated at the parahippocampal / perirhinal border. Moreover, we have softened our language regarding overlap with PPA. The revised text reads as follows:

“Next, we revealed conceptual similarity coding in the conceptual task context within a cluster of voxels that straddled the border between right parahippocampal cortex and PRC (Figure 10B). Although this cluster was only partially situated with parahippocampal cortex, it is interesting to note that its posterior extent did slightly encroach upon anterior aspects of the parahippocampal place area (PPA; functionally defined using a group-level GLM (scenes > objects); Epstein and Kanwisher, 1998), which has previously been linked to the representation of abstract conceptual information (Aminoff et al., 2006; Baldassano et al., 2013; Marchette et al., 2015).”3) Also, there seems to be an error in the caption for Figure 10B, which is described as depicting correlation between brain-based VISUAL and behavior-based conceptual RDMs – both should be conceptual. The label on the figure itself is different (and presumably correct).

We are grateful to the reviewer/Editor for having caught this error. The caption has been revised to indicate that the similarity map depicts significant correlations between brain-based conceptual task RDMs and the behavior-based conceptual RDM.